# Potential Score Matching: Debiasing Molecular Structure Sampling with Potential Energy Guidance

**Liya Guo**[*]                                                                       *gly22@mails.tsinghua.edu.cn*
*Yau Mathematical Sciences Center, Tsinghua University*
*Department of Mathematical Sciences, Tsinghua University*

**Zun Wang**                                                                          *zunwang@microsoft.com*
*Microsoft Research AI4Science, Beijing, China*

**Chang Liu**[†]                                                                      *Chang.Liu@microsoft.com*
*Microsoft Research AI4Science, Beijing, China*

**Junzhe Li**[‡]                                                                      *lijunzhe1028@stu.pku.edu.cn*
*School of Computer Science, Peking University*

**Pipi Hu**[§]                                                                        *pisquare@microsoft.com*
*Microsoft Research AI4Science, Beijing, China*

**Yi Zhu**[¶]                                                                         *yizhu@tsinghua.edu.cn*
*Yau Mathematical Sciences Center, Tsinghua University*
*Yanqi Lake Beijing Institute of Mathematical Sciences and Applications*

**Tao Qin**                                                                           *taoqin@microsoft.com*
*Microsoft Research AI4Science, Beijing, China*

**Reviewed on OpenReview:** *https://openreview.net/forum?id=tTdzbnvTno&invitationId*

## Abstract

The ensemble average of physical properties of molecules is closely related to the distribution of molecular conformations, and sampling such distributions is a fundamental challenge in physics and chemistry. Traditional methods like molecular dynamics (MD) simulations and Markov chain Monte Carlo (MCMC) sampling are commonly used but can be time-consuming and costly. Recently, diffusion models have emerged as efficient alternatives by learning the distribution of training data. Obtaining an unbiased target distribution is still an expensive task, primarily because it requires satisfying ergodicity. To tackle these challenges, we propose Potential Score Matching (PSM), an approach that utilizes the potential energy gradient to guide generative models. PSM does not require exact energy functions and can debias sample distributions even when trained on limited and biased data. Our method outperforms existing state-of-the-art (SOTA) models on the Lennard-Jones (LJ) potential, a commonly used toy model. Furthermore, we extend the evaluation of PSM to high-dimensional problems using the MD17 and MD22 datasets. The results

---

[*]Work done during internship at Microsoft Research AI4Science.
[†]Corresponding author.
[‡]Work done during internship at Microsoft Research AI4Science.
[§]Corresponding author.
[¶]Corresponding author.

demonstrate that molecular distributions generated by PSM more closely approximate the Boltzmann distribution compared to traditional diffusion models.

## 1 Introduction

Physical quantities of interest, such as free energy, are often determined by ensemble averages and are intrinsically linked to the distribution of molecular conformations Alder & Wainwright (1959); Stoltz et al. (2010). Traditional techniques for quantifying these physical quantities, notably Markov Chain Monte Carlo (MCMC) sampling and Molecular Dynamics (MD), are well-established Gelman & Rubin (1996); McCammon et al. (1977); Noé et al. (2019). However, these methods are computationally intensive, particularly for systems with high-dimensional molecules.

Unlike the inherently sequential sampling of MD and MCMC, generative models, especially diffusion models, have emerged as efficient alternatives for generating independent and identically distributed (i.i.d.) samples Song et al. (2020b); Ho et al. (2020); Song et al. (2020a); Phillips et al. (2024); De Bortoli et al. (2022); Wu & Li (2023); Xu et al. (2022); Hoogeboom et al. (2022); Woo & Ahn (2024). One such approach is Denoising Score Matching (DSM) Song et al. (2020b). These models apply a score function to learn the data distribution, which is particularly useful in molecular systems where the equilibrium distribution adheres to the Boltzmann distribution. This distribution can be expressed as $p(\boldsymbol{x}) \propto e^{-\mathcal{E}(\boldsymbol{x})/(k_B T)}$, with $k_B$ denoting the Boltzmann constant, $T$ the temperature, and $\mathcal{E}(\boldsymbol{x})$ the energy function of the system. Constructing a dataset that follows the Boltzmann distribution remains a challenging task, and failure to provide unbiased training data can result in DSM overfitting to a biased distribution, leading to inaccurate observables.

Recent advances in generative modeling have sought to incorporate the principles of the Boltzmann distribution for improved molecular sampling Wu et al. (2024); Woo & Ahn (2024); Bortoli et al. (2024); Chen et al. (2024); Chung et al. (2023). Techniques such as the Denoising Diffusion Sampler (DDS) and the Path Integral Sampler (PIS) Zhang & Chen (2022); Vargas et al. (2023) have been developed to amortize the computational expense of traditional MCMC and MD methods, facilitating learning processes that do not necessitate equilibrium data. These methods typically rely on integration paths derived from ordinary differential equations (ODEs) and stochastic differential equations (SDEs), which, despite being innovative, still involve substantial computational time. The Iterated Denoising Energy Matching (iDEM) framework Akhound-Sadegh et al. (2024) represents another stride forward, employing energy functions for data generation and introducing energy-guided sampling independent of the initial distribution. The complexity of its iterative loops and the absence of predefined initial data increase the computational overhead in high dimensional space, and at larger time scales, iDEM's sampling efficiency diminishes, demanding a greater number of samples to obtain precise outcomes. A more recent approach, Target Score Matching (TSM) Bortoli et al. (2024), integrates energy information with DSM to enable sampling from designated energy functions under the assumption of data unbiasedness.

Table 1: Comparison of molecular sampling methods and their properties.

| | Inference i.i.d. Samples | Doesn't Require Exact Boltzmann Data for Training | Efficiency in High Dim. | Doesn't Require Energy Function (Only Energy Labels) |
|---|---|---|---|---|
| **MD/MCMC** | ✗ | ✓ | ✗ | ✗ |
| **DSM** | ✓ | ✗ | ✓ | ✓ |
| **DDS/PIS** | ✓ | ✗ | ✗ | ✗ |
| **iDEM** | ✓ | ✓ | ✗ | ✗ |
| **TSM** | ✓ | ✗ | ✓ | ✓ |
| **PSM (ours)** | ✓ | ✓ | ✓ | ✓ |

In this work, we introduce the Potential Score Matching (PSM) method, which incorporates potential energy derivatives into generative models to more closely align the sample distribution with the Boltzmann distribution. Furthermore, PSM requires only force labels for the reference structures and obviates the need for an

explicit energy function. PSM leverages the characteristics of DSM to facilitate the generation of i.i.d. samples, offering a computationally efficient alternative to MD simulations. We provide a comparative analysis of PSM with related methodologies in Table 1. We also present a series of theoretical proofs demonstrating that PSM provides a more accurate estimation of the score function in the vicinity of $t = 0$ when training data is biased and results in reduced variance near $t = 0$. The performance of PSM is validated on both simple toy models, such as Lennard-Jones (LJ) potentials, and more complex, high-dimensional physical datasets, including MD17 and MD22. Our experimental results consistently indicate that PSM outperforms baseline models.

## 2 Preliminaries

### 2.1 Score-Based Generative Modeling

Diffusion models operate through a two-step process Ho et al. (2020); Song et al. (2020a): (1) a forward diffusion process that incrementally adds noise to the data $\boldsymbol{x}_0$ until it converges to a Gaussian distribution at time $T$, and (2) a reverse denoising process that reconstructs samples using a learned score function. The forward process is defined by $\boldsymbol{x}_t = \alpha_t \boldsymbol{x}_0 + \sigma_t \boldsymbol{\epsilon}$, where $\alpha_t$ and $\sigma_t$ are noise schedule coefficients, and $\boldsymbol{\epsilon}$ represents standard Gaussian noise. Two widely used frameworks are the variance-preserving (VP-SDE) and variance-exploding (VE-SDE) diffusion processes Song et al. (2020b). In VE-SDE, $\alpha_t = 1$ and $\sigma_t = \sigma_{\min}(\sigma_{\max}/\sigma_{\min})^t$, while in VP-SDE, $\alpha_t = e^{-\frac{1}{2}\int_0^t \beta_s ds}$ and $\sigma_t = \sqrt{1 - e^{-\int_0^t \beta_s ds}}$. Both VP and VE can be abstracted as $\mathrm{d}\boldsymbol{x}_t = f(\boldsymbol{x}_t, t)\, \mathrm{d}t + g(t)\, \mathrm{d}W_t$. Following this, the reverse process, solves the reverse SDE from $T$ back to 0 is

$$\mathrm{d}\boldsymbol{x}_t = \left(f(\boldsymbol{x}_t, t) - g^2(t)\nabla_{\boldsymbol{x}_t}\log q_t\left(\boldsymbol{x}_t\right)\right)\, \mathrm{d}t + g(t)\, \mathrm{d}W_t\,. \tag{1}$$

The term $\nabla_{\boldsymbol{x}_t}\log q_t(\cdot)$, referred to as the "score function" at time $t$, is unknown and is approximated by training a neural network $\boldsymbol{s}_\theta$ parameterized by $\theta$, using the objective function $\|\boldsymbol{s}_\theta - \nabla_{\boldsymbol{x}_t}\log q_t(\cdot)\|_2^2$. However, since the marginal distribution $q_t(\boldsymbol{x}_t)$ is intractable Vincent (2011), later works propose a simplified and equivalent loss function based on the conditional distribution $\|\boldsymbol{s}_\theta - \nabla_{\boldsymbol{x}_t}\log q_t(\boldsymbol{x}_t \mid \boldsymbol{x}_{t-1})\|_2^2$, where the conditional distribution is modeled as a Gaussian transition kernel. In this case, we can leverage the identity $\nabla_{\boldsymbol{x}_t}\log q_t(\boldsymbol{x}_t \mid \boldsymbol{x}_{t-1}) = -\frac{\boldsymbol{\epsilon}}{\sigma_t}$ to instead train a network to predict the noise $\boldsymbol{\epsilon}$ directly. This leads to the commonly used loss function:

$$\min_\theta \mathbb{E}_{t,\boldsymbol{x}_t,\boldsymbol{\epsilon}}\left[\lambda(t)\left\|\boldsymbol{\epsilon}_\theta\left(\boldsymbol{x}_t, t\right) + \sigma_t\nabla_{\boldsymbol{x}_t}\log q_t\left(\boldsymbol{x}_t\right)\right\|_2^2\right] \Leftrightarrow \min_\theta \mathbb{E}_{t,\boldsymbol{x}_t,\boldsymbol{\epsilon}}\left[\lambda(t)\left\|\boldsymbol{\epsilon}_\theta\left(\boldsymbol{x}_t, t\right) - \boldsymbol{\epsilon}\right\|_2^2\right], \tag{2}$$

where the expectation $\mathbb{E}$ is taken over time $t$, sampled from a uniform distribution $\mathcal{U}([0,1])$, the noised data points $\boldsymbol{x}_t$, sampled from the distribution $q(\boldsymbol{x}_t|\boldsymbol{x}_0)$, and the noise $\boldsymbol{\epsilon}$, sampled from a standard Gaussian distribution. The function $\lambda(t)$ represents a weighting function that adjusts the importance of different time steps during the optimization process. In molecular systems, $\boldsymbol{x}_t$ often denotes atomic positions or, in some contexts, discrete atomic types. Diffusion models have become increasingly prevalent in the generation of molecular structures and the prediction of their properties De Bortoli et al. (2022).

### 2.2 Boltzmann Distribution

Molecular systems at equilibrium are typically characterized by the Boltzmann distribution, with the target distribution given by $p(\boldsymbol{x}_0) \propto e^{-\mathcal{E}(\boldsymbol{x}_0)/(k_B T)}$. A profound connection exists between the score function utilized in diffusion models and the system's potential energy, as $\nabla\log p(\boldsymbol{x}_0) \propto -\nabla\mathcal{E}(\boldsymbol{x}_0)/(k_B T)$. Nevertheless, harnessing the Boltzmann distribution for generative modeling poses several challenges. For molecular systems with analytically specified energy functions, the exact distribution is the normalized form of the Boltzmann factor. The normalization constant, typically expressed as an integral over all possible configurations, is often intractable and eludes a closed-form expression. Recent research has integrated energy considerations into score-based model variants to improve molecular predictions. Innovations include the introduction of an equivariant energy-guided SDE Bao et al. (2022) and novel score functions Janner et al. (2022); Durumeric

et al. (2024); Phillips et al. (2024); Wang et al. (2024); Akhound-Sadegh et al. (2024). Additionally, attaining equilibrium presupposes ergodicity in the system, which is ensured by simulating molecular trajectories over extended periods. This requirement poses computational and temporal demands. In our approach, we address these issues by harnessing the derivatives of energy to facilitate this process.

## 3 Potential Score Matching in Molecular Systems

In this section, we introduce our Potential Score Matching (PSM) method, which leverages the derivatives of potential energy to efficiently approximate the Boltzmann distribution. Consider a training sample $\boldsymbol{x}_0$ drawn from the dataset, subjected to a noise injection process defined as $\boldsymbol{x}_t = \alpha_t \boldsymbol{x}_0 + \sigma_t \boldsymbol{\epsilon}$, where $\sigma_t$ is a time-dependent noise schedule. We formalize the representation of the score function for molecules that adhere to the Boltzmann distribution as follows:

**Theorem 1.** *In a molecular system where molecules follow the distribution $p(\boldsymbol{x}_0) \propto e^{-\mathcal{E}/(k_B T)}$, given $\boldsymbol{x}_t = \alpha_t \boldsymbol{x}_0 + \sigma_t \boldsymbol{\epsilon}$, the score function at time $t$ is an expectation of the force,*

$$\nabla_{\boldsymbol{x}_t} \log p(\boldsymbol{x}_t) = \frac{1}{\alpha_t} \mathbb{E}_{\boldsymbol{x}_0|\boldsymbol{x}_t} \left[ \frac{-\nabla_{\boldsymbol{x}_0} \mathcal{E}}{k_B T} \right] = \frac{1}{\alpha_t} \mathbb{E}_{\boldsymbol{x}_0|\boldsymbol{x}_t} \left[ \frac{\boldsymbol{F}}{k_B T} \right], \tag{3}$$

*where $\mathcal{E}$ and $\boldsymbol{F}$ represent the potential energy and force of $\boldsymbol{x}_0$, respectively. Thus, the PSM loss is defined as*

$$\begin{aligned}
\mathcal{L}_{s\text{-}model} &= \mathbb{E}_{t \sim \mathcal{U}(0,1)} \mathbb{E}_{\boldsymbol{x}_t \sim p(\boldsymbol{x}_t|\boldsymbol{x}_0)} \mathbb{E}_{\boldsymbol{x}_0} \left[ \lambda(t) \left\| \boldsymbol{s}_\theta(\boldsymbol{x}_t, t) - \frac{1}{\alpha_t} \left( -\frac{\nabla_{\boldsymbol{x}_0} \mathcal{E}}{k_B T} \right) \right\|^2 \right] \\
&= \mathbb{E}_{t \sim \mathcal{U}(0,1)} \mathbb{E}_{\boldsymbol{x}_t \sim p(\boldsymbol{x}_t|\boldsymbol{x}_0)} \mathbb{E}_{\boldsymbol{x}_0} \left[ \lambda(t) \left\| \boldsymbol{s}_\theta(\boldsymbol{x}_t, t) - \frac{1}{\alpha_t} \frac{\boldsymbol{F}}{k_B T} \right\|^2 \right],
\end{aligned} \tag{4}$$

*where $\mathcal{U}(0,1)$ denotes a uniform distribution, and $p(\boldsymbol{x}_t|\boldsymbol{x}_0)$ is the conditional probability distribution of the noise-injected sample given the $\boldsymbol{x}_0$.*

The proof of Theorem 1 is provided in Appendix B.1. The loss function equation 4, also known as the "score loss", can be further expressed in terms of the "$\boldsymbol{x}_0$ loss" and "$\epsilon$ loss", as detailed in Luo (2022), and the relationship among these three losses is explained in Appendix B.2.

$$\mathcal{L}_{\boldsymbol{x}_0\text{-model}} = \mathbb{E}_{t \sim \mathcal{U}(0,1)} \mathbb{E}_{\boldsymbol{x}_t \sim q(\boldsymbol{x}_t|\boldsymbol{x}_0)} \mathbb{E}_{\boldsymbol{x}_0} \left[ \lambda(t) \left\| \mathcal{D}_\theta - \boldsymbol{x}_t - \frac{\sigma_t^2}{\alpha_t} \frac{\nabla_{\boldsymbol{x}_0} \mathcal{E}(\boldsymbol{x}_0)}{k_B T} \right\|_2^2 \right], \tag{5}$$

$$\mathcal{L}_{\boldsymbol{\epsilon}\text{-model}} = \mathbb{E}_{t \sim \mathcal{U}(0,1)} \mathbb{E}_{\boldsymbol{x}_t \sim q(\boldsymbol{x}_t|\boldsymbol{x}_0)} \mathbb{E}_{\boldsymbol{x}_0} \left[ \lambda(t) \left\| \boldsymbol{\epsilon}_\theta(\boldsymbol{x}_t, t) + \frac{\sigma_t}{\alpha_t} \frac{\boldsymbol{F}}{k_B T} \right\|_2^2 \right], \tag{6}$$

where $\mathcal{D}_\theta$ and $\boldsymbol{\epsilon}_\theta$ are neural networks designed to approximate the original data point $\boldsymbol{x}_0$ and the noise term $\boldsymbol{\epsilon}$, respectively. Recall that the formulas for the VESDE forward process that $\alpha_t = 1$, the loss can be written as $\mathcal{L} = \mathbb{E}_{t \sim U(0,1)} \mathbb{E}_{\boldsymbol{x}_t \sim p(\boldsymbol{x}_t|\boldsymbol{x}_0)} \mathbb{E}_{\boldsymbol{x}_0} \left[ \lambda(t) \| \boldsymbol{s}_\theta(\boldsymbol{x}_t, t) - \frac{\boldsymbol{F}}{k_B T} \|^2 \right]$. For simplicity, we denote $k_B T = 1$ in this section.

Related ideas are explored in Target Score Matching (TSM) Bortoli et al. (2024), which utilizes an explicit energy function to model the score. Building on this concept, we link energy derivatives to the score function and present PSM, which embodies the aforementioned score representation. By utilizing force labels as a proxy to the Boltzmann distribution, our approach reduces computational costs. Our method extends TSM from toy models with analytically defined energy functions to realistic, high-dimensional molecular systems. Unlike prior approaches that rely on energy-based MLPs, we incorporate invariance-preserving architectures with a novel time embedding tailored for molecular data, later introduced in the the experimental section. Moreover, our theoretical findings assert that PSM remains robust even when trained on data that does not adhere to the Boltzmann distribution, thereby correcting biases in the training dataset and yielding samples that more closely resemble the true distribution.

Now we denote that $p(\boldsymbol{x}_0)$ is the Boltzmann distribution and $q(\boldsymbol{x}_0)$ is the data distribution. Ideally, $p(\boldsymbol{x}_0)$ should coincide with $q(\boldsymbol{x}_0)$; however, when starting with a biased distribution, these two may not be equal.

We demonstrate that PSM can provide a superior training process in terms of data fidelity, regardless of whether the training data is biased or unbiased. Since $\nabla_{\boldsymbol{x}_t} \log p(\boldsymbol{x}_t) = \int p(\boldsymbol{x}_0|\boldsymbol{x}_t)\nabla_{\boldsymbol{x}_0} \log p(\boldsymbol{x}_0)d\boldsymbol{x}_0$, the optimal solution satisfies:

$$\underset{\boldsymbol{s}(\cdot,\cdot)}{\arg\min}\, \mathbb{E}_{p(\boldsymbol{x}_0)}\mathbb{E}_{p(\boldsymbol{x}_t|\boldsymbol{x}_0)}\|\boldsymbol{s}(\boldsymbol{x}_t, t) - \nabla_{\boldsymbol{x}_t} \log p(\boldsymbol{x}_t)\| = \mathbb{E}_{p(\boldsymbol{x}_0|\boldsymbol{x}_t)}\left[\nabla_{\boldsymbol{x}_0} \log p(\boldsymbol{x}_0)\right]. \tag{7}$$

Thus, the ground truth is given by $\mathbb{E}_{p(\boldsymbol{x}_0|\boldsymbol{x}_t)}\left[\nabla \log p\left(\boldsymbol{x}_0\right)\right]$, and the Denoising Score Matching (DSM) method learns this same expectation. The following theorem establishes that at small time steps, PSM more closely approximates the Boltzmann distribution than DSM. A detailed explanation is provided in Appendix B.3.

**Theorem 2.** *For the Boltzmann distribution $p(\boldsymbol{x}_0)$, the data distribution $q(\boldsymbol{x}_0)$, and small $t$, PSM learns a more accurate score function than DSM,*

$$\left\|\underbrace{\mathbb{E}_{q(\boldsymbol{x}_0|\boldsymbol{x}_t)}\left[\nabla \log p\left(\boldsymbol{x}_0\right)\right]}_{\text{PSM}} - \underbrace{\mathbb{E}_{p(\boldsymbol{x}_0|\boldsymbol{x}_t)}\left[\nabla \log p\left(\boldsymbol{x}_0\right)\right]}_{\text{Ground Truth}}\right\|_2^2 \leqslant \left\|\underbrace{\mathbb{E}_{q(\boldsymbol{x}_0|\boldsymbol{x}_t)}\left[\nabla \log q\left(\boldsymbol{x}_0\right)\right]}_{\text{DSM}} - \underbrace{\mathbb{E}_{p(\boldsymbol{x}_0|\boldsymbol{x}_t)}\left[\nabla \log p\left(\boldsymbol{x}_0\right)\right]}_{\text{Ground Truth}}\right\|_2^2. \tag{8}$$

*Furthermore, denoting the right-hand side as $\|I_1\|_2^2$ and the left-hand side as $\|I_2\|_2^2$, the difference between these two terms satisfies $\|I_2\|_2^2 - \|I_1\|_2^2 \geq \|I_3\|_2^2 + O(t^3)$, where $I_3 = I_3(\boldsymbol{x}_0, \boldsymbol{x}_t) > 0$ in a neighborhood of $t = 0$.*

Recent works Yang et al. (2023); Phillips et al. (2024) have indicated that during the training of traditional diffusion models, there is a tendency to encounter large Lipschitz constants and significant variance in relation to the time variable near $t = 0$, can be seen in Lemma 2 and 3 in Appendix Appendix B.4, which have the potential to destabilize the training process. These highlight the need for improved training strategies in the small-$t$ regime. Encouragingly, Theorem 2 shows that PSM offers a theoretically grounded debiasing mechanism by promoting convergence toward the Boltzmann distribution in molecular systems. This suggests the potential of applying PSM labels in the small-$t$ region of diffusion models. We propose a weighted combination of both losses, referred to as "Piecewise Loss" and "Piecewise Weighted Loss". These methods prioritize PSM for small $t$ while favoring DSM at larger time values. We denote the noise adding time range of PSM loss as $t_{\text{psm}}$, and similarly denote the time of DSM label as $t_{\text{dsm}}$.

**Piecewise loss.** Assuming $k_B T = 1$ and given a chosen time point $t_p$, we use the force loss (equation 6) exclusively for $t \in [0, t_p]$, while employing DSM for $t > t_p$ as follows. Here, $t_{\text{psm}} \triangleq [0, t_p]$ and $t_{\text{dsm}} \triangleq [t_p, 1]$.

$$\boldsymbol{s}\left(\boldsymbol{x}_t, t\right) = \begin{cases} \frac{1}{\alpha_t}\mathbb{E}_{\boldsymbol{x}_0|\boldsymbol{x}_t}\left[\nabla_{\boldsymbol{x}_0} \log p\left(\boldsymbol{x}_0\right)\right], & \text{if } t < t_p, \\ \mathbb{E}_{\boldsymbol{x}_0|\boldsymbol{x}_t}\left[\nabla_{\boldsymbol{x}_t} \log p\left(\boldsymbol{x}_t \mid \boldsymbol{x}_0\right)\right], & \text{if } t \in [t_p, 1]. \end{cases} \tag{9}$$

**Piecewise Weighted loss.** Since $\boldsymbol{s}\left(\boldsymbol{x}_t, t\right) = \mathbb{E}_{\boldsymbol{x}_0|\boldsymbol{x}_t}\left[\frac{1}{\alpha_t}\nabla_{\boldsymbol{x}_0} \log p\left(\boldsymbol{x}_0\right)\right] = \mathbb{E}_{\boldsymbol{x}_0|\boldsymbol{x}_t}\left[\nabla_{\boldsymbol{x}_t} \log p\left(\boldsymbol{x}_t \mid \boldsymbol{x}_0\right)\right] = \mathbb{E}_{\boldsymbol{x}_0|\boldsymbol{x}_t}\left[\frac{\boldsymbol{x}_0 - \boldsymbol{x}_t}{\sigma_t^2}\right]$, we consider another form of the score

$$s\left(\boldsymbol{x}_t, t\right) = E_{\boldsymbol{x}_0|\boldsymbol{x}_t}\left[\omega_t \frac{1}{\alpha_t}\nabla_{\boldsymbol{x}_0} \log p\left(\boldsymbol{x}_0\right) + (1 - \omega_t)\frac{\boldsymbol{x}_0 - \boldsymbol{x}_t}{\sigma_t^2}\right], \tag{10}$$

and the the loss is $\mathcal{L} = \mathbb{E}_{t,\boldsymbol{x}_t,\boldsymbol{x}_0}\left[\lambda(t)\left\|-\frac{\boldsymbol{\epsilon}_\theta(\boldsymbol{x}_t,t)}{\sigma_t} + \frac{\omega_t}{\alpha_t}\boldsymbol{F} - (1 - \omega_t)\frac{\boldsymbol{x}_0 - \boldsymbol{x}_t}{\sigma_t^2}\right\|_2^2\right]$. We choose $\omega_t$ as a time-varying function which satisfies that $\omega_0 = 1$ and $\omega_1 = 0$. For example, $\omega_t = \text{sigmoid}(50(t - 0.05))$, $t \in [0, 0.1]$ and $\omega_t = 0$ when $t \in [0.1, 1]$. We can see that this loss is a generalization of "Piecewise loss". With a random noise $\boldsymbol{\epsilon}$, the "Piecewise loss" and the "Piecewise Weighted loss" can be unified as

$$\mathcal{L}_{\text{PSM}} = \mathbb{E}_{t,\boldsymbol{x}_t,\boldsymbol{x}_0}\left[\|\boldsymbol{\epsilon}_\theta(\boldsymbol{x}_t, t) - ((1 - \omega_t)\boldsymbol{\epsilon}_1 + \omega_t\boldsymbol{\epsilon}_2)\|_2^2\right], \text{ where } \boldsymbol{\epsilon}_1 = \boldsymbol{\epsilon}[t_{\text{dsm}}],\ \boldsymbol{\epsilon}_2 = -\boldsymbol{F}[t_{\text{psm}}] \cdot \sigma_t,\ t_{\text{dsm}} \in [0, 1] \setminus \{t_{\text{psm}}\}, \tag{11}$$

where $\boldsymbol{\epsilon}$ is a random noise, $t_{\text{psm}}$ refers to the noise adding time range using $\boldsymbol{F}$ as label, and $t_{\text{dsm}}$ is the element in its complement. The training process is shown in Algorithm 1.

---

**Algorithm 1** Potential Score Matching

---

**Require:** Network $\epsilon_\theta$, total iteration $N$, data $\boldsymbol{x}_0$, forces $\boldsymbol{F}$, weight function $\omega_t$, noise schedule $\sigma_t$, and $\alpha_t$;
  Select random noise $\boldsymbol{\epsilon}$;
  Determine the time when the labels $\boldsymbol{F}$ and random noise $\boldsymbol{\epsilon}$ act, $t_{\mathrm{psm}}$ and $t_{\mathrm{dsm}}$.
  **while** $n \leq N$ **do**
    $t \sim \mathcal{U}(0,1)$, $\boldsymbol{x}_t \sim \mathcal{N}(\alpha_t \boldsymbol{x}_0, \sigma_t^2)$, $\boldsymbol{\epsilon}_1 = \boldsymbol{\epsilon}[t_{\mathrm{dsm}}]$, $\boldsymbol{\epsilon}_2 = -\frac{\boldsymbol{F}[t_{\mathrm{psm}}]}{\alpha_t}\sigma_t$;
    $\mathcal{L}_{\mathrm{PSM}} = \|\boldsymbol{\epsilon}_\theta(\boldsymbol{x}_t, t) - ((1-\omega_t)\boldsymbol{\epsilon}_1 + \omega_t\boldsymbol{\epsilon}_2)\|_2^2$;
    $\theta \leftarrow \mathrm{Update}(\theta, \nabla_\theta \mathcal{L}_{\mathrm{PSM}})$;
  **end while**

---

Our approach to incorporating force information is designed to be sample-efficient and does not impose constraints on the sampling methodology. It is compatible with common techniques employed in diffusion models, including the Euler method, Prediction-Correction (PC) method, and EDM Song et al. (2020b); Karras et al. (2022).

## 4 Experiments

**Datasets.** We assess the performance of our proposed Potential Score Matching loss across various settings, including toy models like the Lennard-Jones (LJ) potential Köhler et al. (2020), and more intricate molecular systems such as MD17 and MD22. The LJ configurations, LJ-13 and LJ-55, consist of 13 and 55 atoms, respectively, arranged in a three-dimensional space. Historically, energy-based molecular sampling has largely concentrated on toy models Midgley et al. (2022); Akhound-Sadegh et al. (2024); Woo & Ahn (2024); Bortoli et al. (2024), but our approach extends this to higher-dimensional, real-world datasets, such as MD17 and MD22. MD17 contains molecular trajectories for different molecules, with 9 to 21 atoms, sampled at 500 K, where the thermal energy $k_B T$ in the Boltzmann distribution equals 1 kcal/mol. In contrast, MD22 poses a more demanding challenge due to its greater complexity, featuring molecular dynamics (MD) trajectories from a small peptide with 42 atoms to a double-walled nanotube with 370 atoms Chmiela et al., sampled at 400 to 500 K with a time resolution of 1 fs.

To demonstrate our method's capability to correct biased training distributions and ensembles, we use Denoising Score Matching (DSM) Song et al. (2020b) as a baseline and compare our results with those from iDEM and Flow AIS bootstrap (FAB) in toy model contexts Akhound-Sadegh et al. (2024); Midgley et al. (2022). Our experiments primarily utilize biased training sets, employing the first $1,000$ frames for LJ data and the first $5,000$ frames for MD reference trajectories. Further investigation into the impact of dataset bias is conducted in the ablation study (Section 4.2).

**Diffusion settings and network.** We evaluate our method using DSM, Piecewise, and Piecewise Weighted approaches, employing VESDE for denoising with equal time weight functions, $\lambda(t) = 1$. In the Piecewise approach, PSM is applied for $t \leq 0.05$, while DSM is used for $t \in [0.05, 1]$. In the Piecewise Weighted setting, the label is combined as $\omega_t \times \mathrm{psm\ target} + (1-\omega_t) \times \mathrm{dsm\ target}$, where $\omega_t = \frac{1}{1+\exp(50(t-0.05))}$. During training, we employ the "$\epsilon$-model" loss function to learn the noise $\epsilon$.

Molecular systems often represent molecular structures as graphs, with atomic coordinates, numbers, and bonds defining a molecular graph. Symmetry and equivariance in these graph representations are crucial for accurate molecular interactions. We modify the Equiformer-v2 network based on Liao et al. (2024) by adding the time embedding, more details about this network can be seen in Appendix A.2.2.

For sampling process, we use the Prediction-Correction (PC) sampler Song et al. (2020b), where each prediction step are followed by one correction step. Additional experimental details, including hyperparameter settings, are available in Appendix C.

**Evaluation metrics.** To evaluate the physical plausibility of the generated molecular conformations, we assess the stability of molecules in the MD17 dataset, following the method in Fu et al. (2022). A molecule

is deemed *unstable* at time $T$ if:

$$\max_{(i,j)} \left| \|\boldsymbol{x}_i(T) - \boldsymbol{x}_j(T)\| - b_{ij} \right| > 0.5, \tag{12}$$

where $b_{ij}$ is the equilibrium bond length between atoms $i$ and $j$. If no unstable molecules are detected, the dataset is considered stable. Moreover, we evaluate whether generated samples show an unbiased distribution by analyzing the statistical distribution of *interatomic distances* ($h(r)$), defined as:

$$h(r) = \frac{1}{N(N-1)} \sum_{i=1}^{N} \sum_{j \neq i}^{N} \delta(r - \|\boldsymbol{x}_i - \boldsymbol{x}_j\|), \tag{13}$$

where $r$ is the interatomic distance and $\delta$ is the Dirac delta function. This metric indicates whether the generated samples follow an unbiased molecular distribution. The mean absolute error (MAE) between the sampled $h(r)$ and reference data $h(r)$ also serves as a numerical comparison, providing more intuitive results.

For further numerical insights, we provide the Total Variation Distance (TVD) to measure the maximum distribution distance between sampled and reference data. For probability distributions $P$ and $Q$, TVD is defined as:

$$\mathrm{TVD}(P, Q) = \frac{1}{2} \int_{\mathbb{R}^d} |P(\boldsymbol{x}) - Q(\boldsymbol{x})| \, d\boldsymbol{x},$$

or equivalently for discrete distributions: $\mathrm{TVD}(P, Q) = \frac{1}{2} \sum_{\boldsymbol{x} \in \mathcal{X}} |P(\boldsymbol{x}) - Q(\boldsymbol{x})|$. For datasets with available analytical energy expressions, we additionally provide energy distribution visualizations.

### 4.1 Main Results

#### 4.1.1 Lennard-Jones Potential

The Lennard-Jones (LJ) potential encapsulates the fundamental principles of interatomic interactions: repulsive forces dominate at short distances, while attractive forces prevail at longer ranges. It is mathematically expressed as:

$$\mathcal{E}^{\mathrm{LJ}}(\boldsymbol{x}) = \frac{1}{2\tau} \sum_{ij} \left( \left( \frac{r_m}{d_{ij}} \right)^{12} - 2 \left( \frac{r_m}{d_{ij}} \right)^{6} \right). \tag{14}$$

$$\mathcal{E}^{\mathrm{osc}}(\boldsymbol{x}) = \frac{1}{2} \sum_{i} \|\boldsymbol{x}_i - \boldsymbol{x}_{\mathrm{mean}}\|^2 \quad \mathcal{E}^{\mathrm{tot}} = \mathcal{E}^{\mathrm{LJ}}(\boldsymbol{x}) + \mathcal{E}^{\mathrm{osc}}(\boldsymbol{x}). \tag{15}$$

Here, $r_m$ and $\tau$ are constants that characterize the potential. In our experiments, we use the standard values $r_m = 1$ and $\tau = 1$, in line with previous studies Köhler et al. (2020); Akhound-Sadegh et al. (2024). The term $\mathcal{E}^{\mathrm{osc}}$ represents the harmonic potential energy associated with particle displacements relative to the system's center of mass, $\boldsymbol{x}_{\mathrm{mean}}$.

As indicated by equation 14, the function magnitude significantly increases when any interatomic distance $d_{ij}$ approaches zero, presenting substantial challenges, particularly in high-dimensional systems. To examine the effect of biased training data, we plot the histograms of interatomic distance distributions, $r = d_{ij}$, for LJ-13 and LJ-55, denoted as $h(r)$, in Figure 1. The results demonstrate that PSM effectively debiases samples when trained on biased distributions.

Compared to Akhound-Sadegh et al. (2024); Midgley et al. (2022), we also report the corresponding Wasserstein-2 ($\mathcal{W}$-2) distance and total variation distance (TVD) based on three sets of samples obtained by sampling with different random seeds. Table 2 presents these metric comparisons, where we test the LJ potential using a random 10% subset of the reference data as the training set. Our experimental results show that it is applicable to the toy model and has advantages in high-dimensional situations. From another aspect, PSM also significantly reduces sampling time.

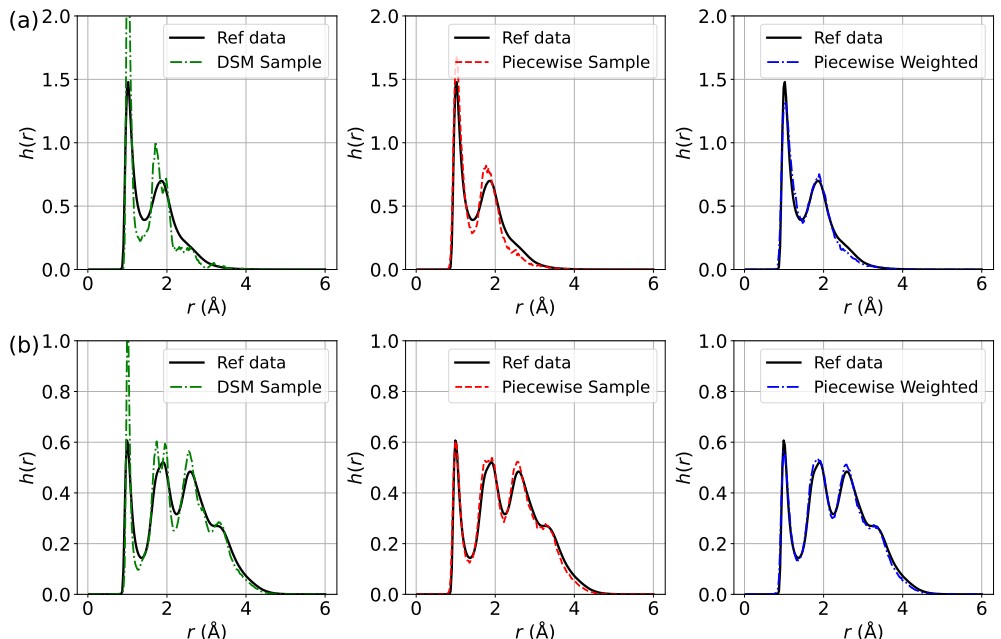

Figure 1: The distribution of interatomic distances $h(r)$ for LJ potential using biased training data with a sample size of 500. (a) Comparison of $h(r)$ for DSM, Piecewise, and Piecewise Weighted losses (from left to right) against reference data for LJ-13; (b) Comparison of $h(r)$ for DSM, Piecewise, and Piecewise Weighted losses for LJ-55.

Table 2: Comparisons of PSM with FAB Midgley et al. (2022) and iDEM Akhound-Sadegh et al. (2024) on LJ-13 and LJ-55 results. Metrics include 2-Wasserstein distance and atomic distance TVD, evaluated on three different seeds.

|  | LJ-13 (39D) | | LJ-55 (165D) | |
| --- | --- | --- | --- | --- |
|  | Sample $\mathcal{W}$-2 | Distance TVD | Sample $\mathcal{W}$-2 | Distance TVD |
| FAB | $4.35 \pm 0.001$ | $0.252 \pm 0.002$ | $18.03 \pm 1.21$ | $0.24 \pm 0.09$ |
| iDEM | $\mathbf{4.26} \pm 0.03$ | $\mathbf{0.044} \pm 0.001$ | $16.128 \pm 0.071$ | $0.09 \pm 0.01$ |
| Piecewise | $4.287 \pm 0.003$ | $\underline{0.0582} \pm 0.001$ | $\mathbf{15.894} \pm \mathbf{0.003}$ | $\underline{0.047} \pm 0.000$ |
| Piecewise-Weighted | $\underline{4.278} \pm 0.001$ | $0.0585 \pm 0.001$ | $\underline{16.054} \pm 0.008$ | $\mathbf{0.023} \pm \mathbf{0.002}$ |

#### 4.1.2 Molecular Dynamical Data

Unlike other studies, our model proves effective on higher-dimensional datasets, specifically testing on MD17 and MD22 datasets. In MD17, we consider molecules such as Uracil (12 atoms), Naphthalene (10 atoms), Aspirin (21 atoms), Salicylic Acid (16 atoms), Malonaldehyde (9 atoms), Ethanol (9 atoms), and Toluene (15 atoms). This dataset provides molecules with properties like Cartesian coordinates (in Å), atomic numbers, total energies (in kcal/mol), and atomic forces (in kcal/mol/Å) at 500 K, where the thermal energy $k_BT$ in the Boltzmann distribution corresponds to 1 kcal/mol. Additionally, we test the MD22 benchmark dataset, which includes Ac-Ala3-NHMe (42 atoms), Docosahexaenoic Acid (DHA) (56 atoms), Stachyose (87 atoms), DNA base pair (AT-AT) (60 atoms), DNA base pair (AT-AT-CG-CG) (118 atoms), Buckyball catcher (148 atoms), and Double-walled nanotube (370 atoms).

**MD17 dataset.** We first evaluate the stability of generated molecular configurations across DSM, Piecewise, and Piecewise Weighted methods, as proposed in Fu et al. (2022). Stability is assessed on $1,000$ samples generated at the epoch with the lowest training loss over $1,000$ epochs. All samples meet the stability criterion. With no unstable molecules in any of the sampled data, we plot the distribution of interatomic distances for these MD17 datasets, as shown in Figure 2. The figure indicates that at the peak, DSM tends to capture

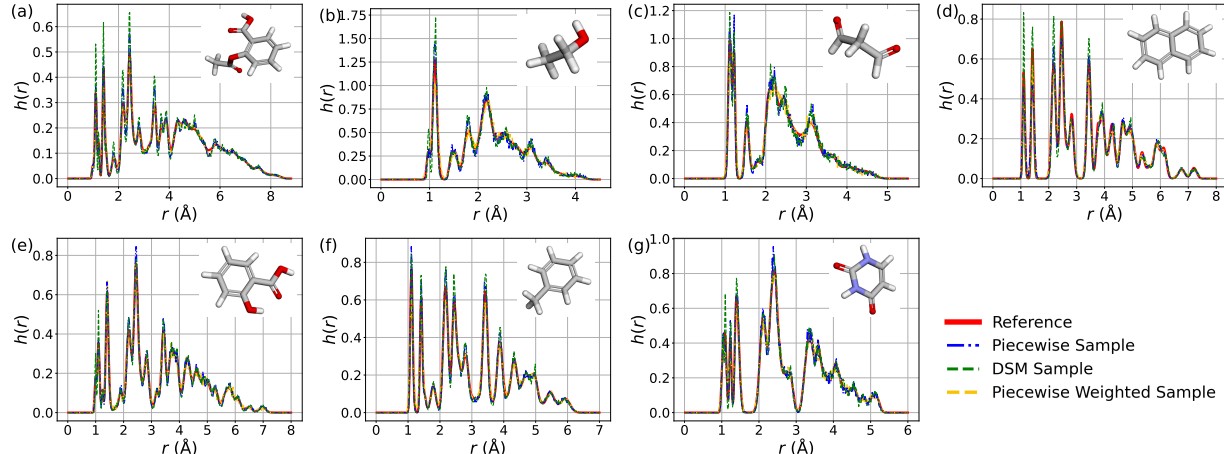

Figure 2: The distribution of interatomic distances $h(r)$ of $r(\text{Å})$ for (a) aspirin, (b) ethanol, (c) malonaldehyde, (d) naphthalene, (e) salicylic acid, (f) toluene, and (g) uracil in MD17 dataset. The insets display the ball-and-stick representations of these molecules learned by Piecewise method. The sample number is $1,000$.

greater fluctuations, and PSM can more accurately learn the neighboring structure. Table 3 summarizes the mean absolute errors (MAEs) of the sampled interatomic distances and total variation distance compared to the reference molecular data. These results demonstrate that, despite the inherent bias in the training data, PSM successfully debiases molecular samples.

Table 3: Comparison of MAE of interatomic distance and Total Variation Distance (TVD) for different MD17 molecules with sample size $1,000$.

| MD 17 | MAE of h(r) | | | TVD | | |
|---|---|---|---|---|---|---|
| | DSM | Piecewise | Piecewise Weighted | DSM | Piecewise | Piecewise Weighted |
| Aspirin | 0.1240 | 0.0570 | **0.0510** | 0.0662 | 0.0373 | **0.0290** |
| Ethanol | 0.1193 | **0.0845** | 0.0911 | 0.0663 | **0.0508** | 0.0501 |
| Malonaldehyde | 0.1003 | 0.0843 | **0.0630** | 0.0515 | **0.0492** | 0.0518 |
| Naphthalene | 0.1257 | **0.0701** | 0.1164 | 0.0716 | **0.0365** | 0.0592 |
| Salicylic Acid | 0.0799 | **0.0554** | 0.0584 | 0.0450 | 0.0424 | **0.0337** |
| Toluene | 0.0970 | **0.0769** | 0.0948 | 0.0536 | **0.0428** | 0.0490 |
| Uracil | 0.0747 | 0.0705 | **0.0560** | 0.0478 | **0.0421** | 0.0426 |

**MD22 dataset.** Similarly, Figure 3 and Table 4 present $h(r)$ and the numerical metrics of the MD22 dataset, showing improved performance with the PSM method. The sample size is 128.

Table 4: Comparison of MAE of $h(r)$ and Total Variation Distance (TVD) for different MD22 molecules.

| MD 22 | MAE of h(r) | | | TVD | | |
|---|---|---|---|---|---|---|
| | DSM | Piecewise | Piecewise Weighted | DSM | Piecewise | Piecewise Weighted |
| Ac-Ala3-NHMe | 0.0873 | 0.0505 | **0.0486** | 0.0415 | **0.0224** | 0.0301 |
| Stachyose | 0.0384 | **0.0333** | 0.0424 | 0.0174 | **0.0161** | 0.0198 |
| Docosahexaenoic acid | 0.0602 | **0.0444** | 0.0518 | 0.0254 | **0.0221** | 0.0228 |
| AT-AT | 0.0767 | 0.0620 | **0.0611** | 0.0336 | 0.0292 | **0.0285** |
| AT-AT-CG-CG | 0.0706 | **0.0655** | 0.0677 | 0.0341 | **0.0324** | 0.0329 |
| Buckyball catcher | 0.0978 | **0.0350** | 0.0567 | 0.0417 | **0.0158** | 0.0264 |
| Dw-nanotube | 0.1527 | 0.0919 | **0.0484** | 0.0946 | 0.0597 | **0.0256** |

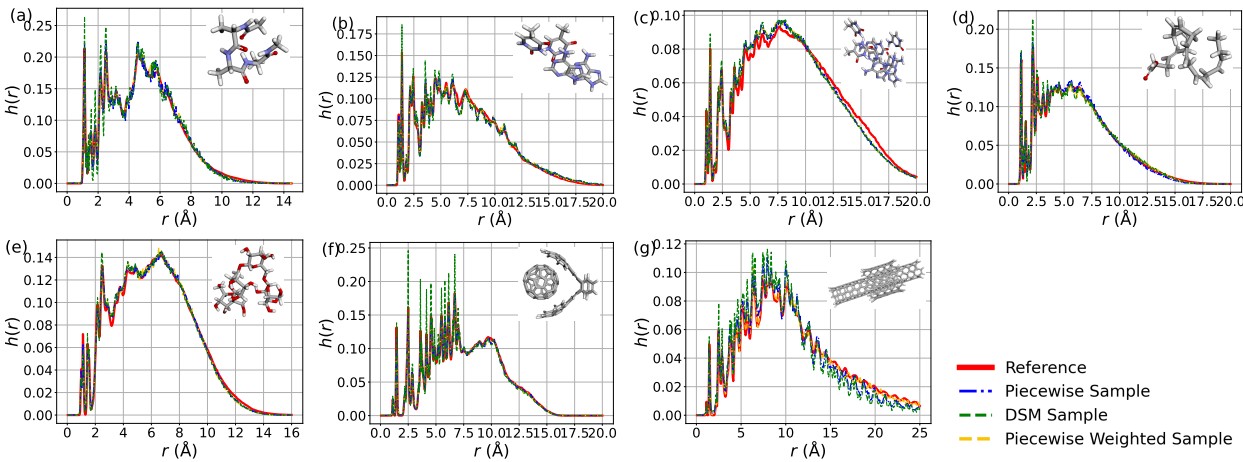

Figure 3: The distribution of interatomic distances ($r$(Å)) for (a) Ac-Ala3-NHMe, (b) DNA base pair (AT-AT), (c) DNA base pair (AT-AT-CG-CG), (d) Docosahexaenoic acid, (e) Stachyose, (f) Buckyball catcher, (g) Dw nanotube in MD22 dataset. The insets display the ball-and-stick representations of these molecules.

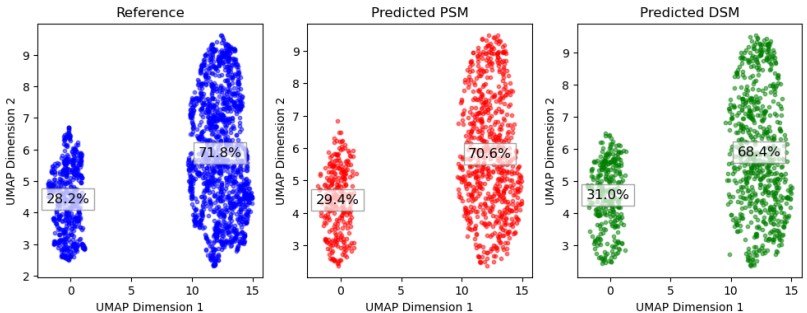

Figure 4: UMAP of SOAP feature for reference data, PSM, and DSM.

**Debiasing conformations using SOAP descriptors.** In molecular dynamics, to illustrate that PSM extends beyond recovering ensemble averages such as $h(r)$ and achieves debiased results in terms of structural accuracy, we utilize the Smooth Overlap of Atomic Positions (SOAP) descriptor. SOAP describes the local atomic environment around a central atom $\alpha$ as a smoothly varying atomic density function:

$$\rho_\alpha(\mathbf{r}) = \sum_i f_c(r_{i\alpha})g(r_{i\alpha})Y_{lm}(\hat{\mathbf{r}}_{i\alpha}), \tag{16}$$

where $r_{i\alpha} = |\mathbf{r}_i - \mathbf{r}_\alpha|$ is the distance between the central atom $\alpha$ and its neighbor $i$. The function $f_c(r)$ ensures locality, $g(r)$ is a radial basis function, and $Y_{lm}(\hat{\mathbf{r}})$ are spherical harmonics describing angular dependence. SOAP maintains a high-dimensional feature representation invariant under rotations. To visualize SOAP results, we apply UMAP, a nonlinear dimensionality reduction technique, for intuitive comparison.

To demonstrate PSM's capability in debiasing, we construct a deliberately biased training dataset informed by our ablation study (Section 4.2), which reveals that the first $1,000$ simulation frames exhibit structural deviation. Specifically, we augment a randomly selected subset (comprising $0.5\%$ of the full trajectory, approximately $1,000$ frames) with these biased early frames to introduce controlled bias into the training data. We then compare the resulting molecular distributions generated by PSM, DSM, and the reference data using UMAP-projected SOAP features clustered via DBSCAN. As shown in Figure 4, the cluster proportions for the reference, PSM, and DSM are $(28.2\%, 71.8\%)$, $(29.4\%, 70.6\%)$, and $(31.0\%, 68.4\%)$, respectively. The close alignment between PSM and the reference indicates that PSM more effectively captures the underlying atomic environments and debiases the molecular distribution.

## 4.2 Ablation Study

In this section, we conduct ablation experiments on the debias capability of PSM and the amortized MD computation cost to illustrate the efficiency of PSM.

**Debias ability.** First, we investigate the impact of biased versus unbiased training data on the performance of DSM and PSM. To emphasize the effect of debiasing, we conduct experiments on both biased and unbiased datasets. Specifically, we use two different training sets: (1) a biased dataset consisting of the first $1,000$ frames of the trajectory, and (2) an unbiased dataset created by randomly selecting 10% of the reference data. Figure 5 (a) confirms that these datasets exhibit distinct bias characteristics. For this study, we use the ethanol molecule as an example.

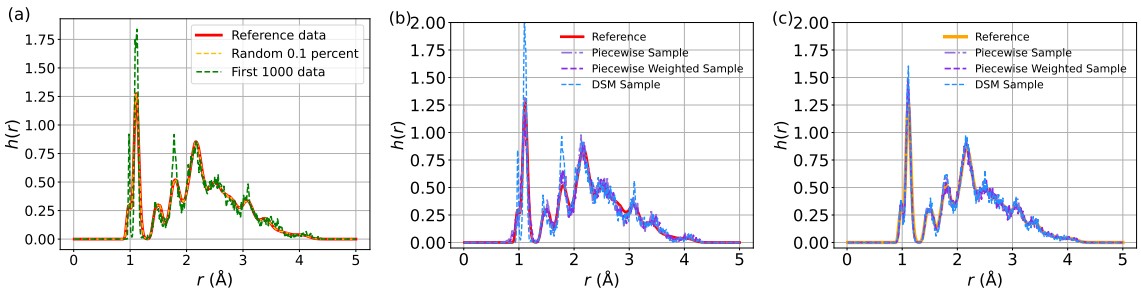

Figure 5: The distribution of interatomic distance comparisons. (a) Distribution of interatomic distances in the reference dataset, the randomly selected subset, and the first $1,000$ frames. (b) Comparison of DSM, Piecewise, and Piecewise Weighted methods trained on the first $1,000$ frames. (c) Comparison of the three methods when trained on a randomly selected 10% subset of the reference data.

Figure 5 (b) compares the sample distributions generated by DSM and PSM when trained on the first 1,000 frames. The results show that DSM mainly reflects the characteristics of the biased training data, whereas PSM successfully corrects the bias, producing samples that better align with the unbiased distribution. In contrast, Figure 5 (c) demonstrates that when the training data is initially unbiased, both DSM and PSM yield accurate estimates. These findings underscore the importance of data quality in DSM training and highlight PSM's capability to generating samples that more accurately approximate the Boltzmann distribution.

**Amortized MD computation cost.** We further compare PSM with molecular dynamics simulations on the aspirin molecule. For MD, we adopt the QuinNet force field Wang et al. (2023), following the original training protocol. We measure both the force field training time and the time required to generate sufficiently long, converged trajectories (treated as the sampling time). For PSM, we train the diffusion model for 500 epochs until convergence and measure the sampling time using the Euler discretization scheme. All experiments are conducted on a single NVIDIA A100 GPU. Our results show that PSM completes training in approximately 8 hours and requires only 10 minutes to generate 1,000 batches, each consisting of 1,000 time steps. In contrast, training the MD force field to a comparable accuracy takes around 24 hours, while achieving state-of-the-art performance may require up to one week of training. Moreover, the MD simulation phase takes about 31 hours and 18 minutes to generate $210,000$ steps—the trajectory length used in the reference MD17 aspirin dataset. These results demonstrate that PSM offers substantially improved sampling efficiency over MD.

## 5 Conclusions

In this study, we introduce Potential Score Matching (PSM), a method designed to mitigate biases in non-Boltzmann distributions by incorporating force labels. Traditional molecular dynamics simulations are computationally expensive. While diffusion models offer an alternative by learning a score function to generate samples that satisfy the training data distribution, obtaining training data that accurately follows

the Boltzmann distribution is challenging. PSM relaxes these constraints. Theoretical analysis indicates that PSM provides training with lower variance and more accurate score estimations in the vicinity of $t = 0$, which allows for a concentrated training effort around this point. This has led to the development of two novel loss formulations: Piecewise and Piecewise Weighted losses. Empirical assessments validate the performance of PSM over Denoising Score Matching by effectively debiasing data towards the Boltzmann distribution. Furthermore, our research extends the application of PSM to high-dimensional real-world datasets, such as MD22 and MD17, demonstrating its capability in handling complex molecular modeling challenges.

Additionally, flow matching has proven to be an effective method for generating molecular conformations and predicting properties Lipman et al. (2022); Chen & Lipman (2023); Miller et al. (2024). Progress in this area Woo & Ahn (2024); Domingo-Enrich et al. (2024) has resulted in unified frameworks that integrate diffusion models and flow matching techniques. Considering the established relation between the velocity field and the score function, as expressed by $v(\boldsymbol{x}, t) = \frac{\dot{\alpha}_t}{\alpha_t} \boldsymbol{x} + \gamma_t \left( \frac{\dot{\alpha}_t}{\alpha_t} \gamma_t - \dot{\gamma}_t \right) s(\boldsymbol{x}, t)$, where $\boldsymbol{x}_t = \alpha_t \boldsymbol{y}_1 + \gamma_t \boldsymbol{y}_0 = \alpha_t \boldsymbol{x}_0 + \gamma_t \varepsilon$, and $\boldsymbol{y}_0$, $\boldsymbol{y}_1$ are sampled from the noise and data distributions respectively, our PSM framework is well-positioned for extension to flow matching. This prospective expansion is earmarked for future exploration, while further theoretical insights and formulations for VPSDE and VESDE are delineated in Appendix D.

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

# A  Preliminary

## A.1  Diffusion Models

A diffusion model typically consists of a forward noise addition process and a reverse denoising process. In the forward process, noise is iteratively added to the data $\boldsymbol{x}$ (at $t = 0$), gradually transforming the distribution into an approximation of a Gaussian distribution at time $T$, independent of the initial state $\boldsymbol{x}_0$. The reverse process then generates samples that conform to the data distribution by leveraging information obtained from the forward process.

We introduce the Variance-Preserving Stochastic Differential Equation (VPSDE) and the Variance-Exploding Stochastic Differential Equation (VESDE) Song et al. (2020b), summarized in the following Table 5. Here, $\mathbf{z}$ is a random noise, $i \in \{1, \cdots, N\}$ is one of noising steps.

Table 5: VESDE and VPSDE.

|  | VESDE | VPSDE |
|---|---|---|
| Noise addition | $\mathbf{x}_i = \mathbf{x}_{i-1} + \sqrt{\sigma_i^2 - \sigma_{i-1}^2}\mathbf{z}_{i-1}$ | $\mathbf{x}_i = \alpha_i\mathbf{x}_0 + \sigma_i\mathbf{z}_i$ |
| SDE Formulation | $d\mathbf{x}_t = \sigma(t)dW_t$ | $d\mathbf{x}_t = -\frac{1}{2}\beta_t\mathbf{x}_t dt + \sqrt{\beta_t}dW_t$ |
| Parameterization | $\sigma(t) = \sigma_{\min}(\sigma_{\max}/\sigma_{\min})^t$ | $\beta_t = \beta_{\min} + (\beta_{\max} - \beta_{\min})t$ |
| Transition Distribution | $\mathcal{N}\left(\boldsymbol{x}_t; \boldsymbol{x}_0, \sigma_{\min}^2(\sigma_{\max}/\sigma_{\min})^{2t}\mathbf{I}\right)$ | $\mathcal{N}\left(\boldsymbol{x}_t; \boldsymbol{x}_0 e^{-\frac{1}{2}\int_0^t \beta_s ds}, (1 - e^{-\int_0^t \beta_s ds})\boldsymbol{I}\right)$ |

We sample from the diffusion model follows a Predictor-Corrector (PC) scheme. The predictor step follows the time-reversed SDE, while the corrector step applies Langevin dynamics to refine the sample. Given a time step $\Delta t$, the predictor step updates the sample as:

$$\tilde{\mathbf{x}}_{t-\Delta t} = \mathbf{x}_t - f(\mathbf{x}_t, t)\Delta t + g(t)\sqrt{\Delta t}\mathbf{z}_t, \tag{17}$$

where $\mathbf{z}_t \sim \mathcal{N}(\mathbf{0}, \mathbf{I})$ is Gaussian noise, and $f$ and $g$ are drift and diffusion coefficients. The corrector step refines $\tilde{\mathbf{x}}_{t-\Delta t}$ using Langevin dynamics $\mathbf{x}_{t-\Delta t} = \tilde{\mathbf{x}}_{t-\Delta t} + \eta\nabla_{\mathbf{x}}\log p_t(\mathbf{x}) + \sqrt{2\eta}\mathbf{z}'$, where $\eta$ is the step size, and $\mathbf{z}' \sim \mathcal{N}(\mathbf{0}, \mathbf{I})$. This two-step process improves the quality of generated samples and enhances convergence to the data distribution.

## A.2 Equiformer-v2

### A.2.1 Equivariance and Irreducible Representation

In molecular systems, molecular structures are typically represented as graph, where atomic coordinates, atomic numbers, and bonded connections define a molecular graph. Ensuring properties such as symmetry and equivariance in these graph representations is crucial for capturing molecular interactions accurately.

**Graph representation.** A molecular graph is defined as $G = (V, E)$, where $V = \{v_1, v_2, \ldots, v_{|V|}\}$ represents the set of atoms (nodes), and $E = \{e_{ij} \mid (i, j) \subset V \times V\}$ denotes the set of edges capturing atomic interactions. The number of atoms in a molecule is denoted as $N = |V|$. Each node $v_i \in V$ is characterized by its nuclear charge $z_i$ and 3D coordinate $r_i \in \mathbb{R}^3$. Our goal is to design a generative model that learns to generate molecular distributions while preserving the underlying chemical and spatial properties.

**Graph neural networks.** Graph Neural Networks (GNNs) are widely used to process graph-structured data by propagating information across nodes and edges. At each iteration $t$, node features $\mathbf{h}_v^{(t)}$ are updated using a message-passing mechanism:

$$\mathbf{m}_v^{(t+1)} = \sum_{u \in \mathcal{N}(v)} \phi(\mathbf{h}_v^{(t)}, \mathbf{h}_u^{(t)}, \mathbf{e}_{vu}), \quad \mathbf{h}_v^{(t+1)} = \psi(\mathbf{h}_v^{(t)}, \mathbf{m}_v^{(t+1)}), \tag{18}$$

where $\mathcal{N}(v)$ represents the neighboring nodes of $v$, $\mathbf{e}_{vu}$ denotes the edge feature between nodes $v$ and $u$, and $\phi(\cdot)$ and $\psi(\cdot)$ are learnable functions, implemented as networks.

**Equivariance and irreducible representations.** Equivariance serves as a fundamental property in neural networks, providing strong prior knowledge that enhances data efficiency in molecular modeling. A function $f : X \to Y$ is equivariant under a transformation group $G$ if, for any input $x \in X$, output $y \in Y$, and group element $g \in G$, $f(D_X(g)x) = D_Y(g)f(x)$ holds, where $D_X(g)$ and $D_Y(g)$ are transformation matrices parameterized by $g$ in $X$ and $Y$. In 3D atomistic graphs, molecular structures must maintain equivariance under the Euclidean group $E(3)$, which includes translations, rotations, and reflections.

For the Euclidean group $E(3)$, scalar quantities remain invariant under rotations, whereas vector quantities transform accordingly. To enforce translation symmetry, computations are performed on relative positions. Since rotation and inversion transformations commute, any representation of the special orthogonal group

$SO(3)$ can be decomposed into irreducible representations (irreps), which serve as fundamental transformation components. Equivariant neural networks leverage these irreducible representations to construct features that remain equivariant to 3D rotations.

For probabilistic models, the equivariance property extends to probability distributions. A probability distribution $p(\boldsymbol{x}_t)$ is equivariant under the special Euclidean group $SE(3)$ if, for any transformation $T_g$ corresponding to $g \in SE(3)$, then $p(\boldsymbol{x}_t) = p(T_g(\boldsymbol{x}_t))$, and $p(\boldsymbol{x}_{t-1} \mid \boldsymbol{x}_t) = p(T_g(\boldsymbol{x}_{t-1}) \mid T_g(\boldsymbol{x}_t))$. A similar definition applies for $E(3)$.

### A.2.2 Equiformer-v2 Network

In our research, we harness the capabilities of the Equiformer-v2 network Liao et al. (2024) to maintain $SE(3)/E(3)$-equivariance, a critical aspect for accurately modeling molecular configurations. This advanced architecture builds on the principles of irreducible representations, ensuring that its operations and features are equivariant and robust. Other works have also incorporated designs ensuring molecular invariance and equivariance Xu et al. (2022); Shi et al. (2021); Le et al. (2022); Musaelian et al. (2023).

Building upon the Equiformer, the Equiformer-v2 introduces sophisticated equivariant graph attention mechanisms, supplanting standard Transformer operations with their SE(3)/E(3)-equivariant counterparts and utilizing tensor products for higher-degree representations. Node embeddings are crafted by concatenating channel-dimension features and applying rotations based on relative positions or edge orientations.

To enhance computational efficiency, Equiformer-v2 forgoes separate depth-wise tensor products and linear layers in favor of a single SO(2) linear layer that processes scalar and irreps features distinctly. It encodes edge distance information using radial basis functions, integrating them into the node embeddings.

The attention mechanisms benefit from extra normalization and separable $S^2$ activation for improved stability. The network's feed-forward section employs a two-layer MLP with SiLU activation. The output head either aggregates scalar predictions or computes atom-wise forces, utilizing equivariant graph attention to boost expressiveness and scalability.

**ESCN convolution.** The ESCN convolution refines equivariant tensor products by substituting conventional SO(3) operations with SO(2) linear ones. Typically, SO(3) convolutions combine input irreps features with spherical harmonics of relative positions using Clebsch-Gordan coefficients. ESCN simplifies this by aligning the relative position vector with a fixed axis through rotation, reducing dependencies and limiting interactions to cases where $m_f = 0$, which streamlines computations while preserving equivariance.

**Attention re-normalization.** To maintain stability at higher $L_{\max}$, we introduce an additional layer normalization step before non-linear transformations, ensuring well-scaled scalar features $f_{ij}^{(0)}$. Attention weights are computed through a leaky ReLU and a linear layer, promoting numerical stability in operations like softmax, which aids in training convergence and model robustness.

**Separable $S^2$ activation** The separable $S^2$ activation processes degree-0 and higher-degree features separately. Degree-0 vectors are partially activated with SiLU, while the remainder is blended with higher-degree vectors for $S^2$ activation. This selective processing minimizes cross-degree interference, stabilizes gradients, and enhances expressiveness, particularly in FFNs.

**Separable layer normalization.** Separable layer normalization (SLN) extends traditional equivariant normalization by independently normalizing degree-0 and higher-degree features. By computing their means and standard deviations separately, SLN preserves inter-degree dynamics, enhancing stability and expressiveness, especially in high $L_{\max}$ scenarios.

Our network architecture is predicated on Equiformer-v2, which yields both one-dimensional energy and three-dimensional force terms. For the $\epsilon$-net output, we utilize only the energy head. Furthermore, we adapt the model to consider time as an additional input feature to align with the diffusion model's time-dependent noise addition.

## B  Derivation of Loss Function

### B.1  Proof of Theorem 1

*Proof.* Assume that $p(\boldsymbol{x}_0) \propto e^{-\mathcal{E}/(k_B T)}$, the derivation of the loss function follows from the transformation of scores using the Gaussian transition probability density function and the integration by parts. Since $\nabla_{\boldsymbol{x}_t} p(\boldsymbol{x}_t|\boldsymbol{x}_0) = \frac{1}{\alpha_t} \nabla_{\boldsymbol{x}_0} p(\boldsymbol{x}_t|\boldsymbol{x}_0)$,

$$
\begin{aligned}
\nabla_{\boldsymbol{x}_t} \log p(\boldsymbol{x}_t) &= \frac{\nabla_{\boldsymbol{x}_t} \int p(\boldsymbol{x}_0) p(\boldsymbol{x}_t|\boldsymbol{x}_0) d\boldsymbol{x}_0}{p(\boldsymbol{x}_t)} = \frac{\int p(\boldsymbol{x}_0) \nabla_{\boldsymbol{x}_t} \left( e^{-\frac{\|\boldsymbol{x}_t - \alpha_t \boldsymbol{x}_0\|^2}{2\sigma_t^2}} \right) d\boldsymbol{x}_0}{p(\boldsymbol{x}_t)} \\
&= \frac{\int p(\boldsymbol{x}_0) \left( -\frac{1}{\alpha_t} \nabla_{\boldsymbol{x}_0} e^{-\frac{\|\boldsymbol{x}_t - \alpha_t \boldsymbol{x}_0\|^2}{2\sigma_t^2}} \right) d\boldsymbol{x}_0}{p(\boldsymbol{x}_t)} = \frac{\int \frac{1}{\alpha_t} \left( \nabla_{\boldsymbol{x}_0} p(\boldsymbol{x}_0) \right) e^{-\frac{\|\boldsymbol{x}_t - \alpha_t \boldsymbol{x}_0\|^2}{2\sigma_t^2}} d\boldsymbol{x}_0}{p(\boldsymbol{x}_t)} \\
&= \int \frac{1}{\alpha_t} \left( p(\boldsymbol{x}_0) \nabla_{\boldsymbol{x}_0} \log p(\boldsymbol{x}_0) \right) \frac{1}{p(\boldsymbol{x}_t)} e^{-\frac{\|\boldsymbol{x}_t - \alpha_t \boldsymbol{x}_0\|^2}{2\sigma_t^2}} d\boldsymbol{x}_0 \\
&= \int \frac{1}{\alpha_t} p(\boldsymbol{x}_0|\boldsymbol{x}_t) \nabla_{\boldsymbol{x}_0} \log p(\boldsymbol{x}_0) d\boldsymbol{x}_0 \\
&= \frac{1}{\alpha_t} \mathbb{E}_{\boldsymbol{x}_0|\boldsymbol{x}_t} \left[ \frac{-\nabla_{\boldsymbol{x}_0} \mathcal{E}}{k_B T} \right] = \frac{1}{\alpha_t} \mathbb{E}_{\boldsymbol{x}_0|\boldsymbol{x}_t} \left[ \frac{\boldsymbol{F}}{k_B T} \right],
\end{aligned}
\tag{19}
$$

where $\mathcal{E}$ is the energy function, and $\boldsymbol{F}$ represents the force derived from $-\nabla_{\boldsymbol{x}_0} \mathcal{E}$. Specifically, for the VESDE, $\alpha_t = 1$, then $\nabla_{\boldsymbol{x}_t} \log p(\boldsymbol{x}_t) = \mathbb{E}_{\boldsymbol{x}_0|\boldsymbol{x}_t} \left[ -\frac{\nabla_{\boldsymbol{x}_0} \mathcal{E}}{k_B T} \right]$.

For simplicity, let $k_B T = 1$. Combining these results, the potential score matching loss can be derived as:

$$
\mathcal{L}_{\text{score - model}} = \mathbb{E}_{t \sim U(0,1)} \mathbb{E}_{\boldsymbol{x}_t \sim p(\boldsymbol{x}_t|\boldsymbol{x}_0)} \mathbb{E}_{\boldsymbol{x}_0} \| s_\theta(\boldsymbol{x}_t, t) - (-\frac{1}{\alpha_t} \nabla_{\boldsymbol{x}_0} \mathcal{E}) \|^2
\tag{20}
$$

$$
= \mathbb{E}_{t \sim U(0,1)} \mathbb{E}_{\boldsymbol{x}_t \sim p(\boldsymbol{x}_t|\boldsymbol{x}_0)} \mathbb{E}_{\boldsymbol{x}_0} \| s_\theta(\boldsymbol{x}_t, t) - \frac{1}{\alpha_t} \boldsymbol{F} \|^2.
\tag{21}
$$

□

The reason why $\nabla_{\boldsymbol{x}_t} \log p(\boldsymbol{x}_t) = \mathbb{E}_{\boldsymbol{x}_0|\boldsymbol{x}_t} [-\nabla_{\boldsymbol{x}_0} \mathcal{E}]$ (objective equation 19) is the solution of the loss function (equation 20) lies in Theorem 3. According to the above expansion, label $\nabla_{\boldsymbol{x}_t} \log p(\boldsymbol{x}_t)$ can be rewritten as

$$
\begin{aligned}
\nabla_{\boldsymbol{x}_t} \log p(\boldsymbol{x}_t) &= \frac{\nabla_{\boldsymbol{x}_t} \int p(\boldsymbol{x}_0) p(\boldsymbol{x}_t|\boldsymbol{x}_0) d\boldsymbol{x}_0}{p(\boldsymbol{x}_t)} = \frac{\int p(\boldsymbol{x}_0) (-\frac{\boldsymbol{x}_t - \boldsymbol{x}_0}{\sigma_t^2}) p(\boldsymbol{x}_t|\boldsymbol{x}_0) d\boldsymbol{x}_0}{p(\boldsymbol{x}_t)} \\
&= \int p(\boldsymbol{x}_0|\boldsymbol{x}_t) (-\frac{\boldsymbol{x}_t - \boldsymbol{x}_0}{\sigma_t^2}) d\boldsymbol{x}_0 = \mathbb{E}_{\boldsymbol{x}_0|\boldsymbol{x}_t} \left[ \frac{\boldsymbol{x}_0 - \boldsymbol{x}_t}{\sigma_t^2} \right],
\end{aligned}
\tag{22}
$$

which is consistent with the writing of "$\boldsymbol{x}_0$-model". Similarly, we can rewrite the loss function with a network representing $\boldsymbol{x}_0$.

$$
\mathbb{E}_{\boldsymbol{x}_0|\boldsymbol{x}_t} \left[ \frac{\boldsymbol{x}_0 - \boldsymbol{x}_t}{\sigma_t^2} + \nabla_{\boldsymbol{x}_0} \mathcal{E}(\boldsymbol{x}_0) \right] = \mathbb{E}_{\boldsymbol{x}_0|\boldsymbol{x}_t} \left[ \frac{\boldsymbol{x}_0 - \boldsymbol{x}_t + \sigma_t^2 \nabla_{\boldsymbol{x}_0} \mathcal{E}(\boldsymbol{x}_0)}{\sigma_t^2} \right] = \mathbb{E}_{\boldsymbol{x}_0|\boldsymbol{x}_t} \left[ \frac{\boldsymbol{x}_0 - \mathcal{D}_\theta}{\sigma_t^2} \right].
\tag{23}
$$

$$
\mathcal{L}_{\boldsymbol{x}_0-\text{model}} = \mathbb{E}_{t \sim U(0,1)} \mathbb{E}_{\boldsymbol{x}_t \sim p(\boldsymbol{x}_t|\boldsymbol{x}_0)} \mathbb{E}_{\boldsymbol{x}_0} \| \mathcal{D}_\theta - \boldsymbol{x}_t - \sigma_t^2 \nabla_{\boldsymbol{x}_0} \mathcal{E}(\boldsymbol{x}_0) \|_2^2.
\tag{24}
$$

**Theorem 3.** *Let $X$ be an integrable random variable. Then for each $\sigma$-algebra $\mathcal{V}$ and $Y \in \mathcal{V}$, $Z = \mathbb{E}(X|\mathcal{V})$ solves the least square problem Evans (2012)*

$$
\|Z - X\| = \min_{Y \in \mathcal{V}} \|Y - X\|,
$$

*where $\|Y\| = \left( \int Y^2 dP \right)^{\frac{1}{2}}$.*

To prove the theorem, we first introduce a lemma.

**Lemma 1.** *If $Y$ is $\mathcal{V}$-measurable, and $f$ is a measurable function in the sense that its domain and codomain are appropriately aligned with the $\sigma$-algebras, then $f(Y)$ will also be $\mathcal{V}$-measurable Evans (2018).*

**Proof of Theorem 3.** Let $X$ be an integrable random variable, $\mathcal{V}$ be a $\sigma$-algebra, and $Y \in \mathcal{V}$. We need to show that $Z = \mathbb{E}(X|\mathcal{V})$ minimizes the least square problem $|Z - X| = \min_{Y \in \mathcal{V}} |Y - X|$.

First, recall the property of conditional expectation: $\mathbb{E}(Z|\mathcal{V}) = Z$ for any $Z \in \mathcal{V}$. Consider the difference $X - Y$ for any $Y \in \mathcal{V}$. We can write this difference as:

$$X - Y = (X - \mathbb{E}(X|\mathcal{V})) + (\mathbb{E}(X|\mathcal{V}) - Y)$$

Note that $\mathbb{E}(X|\mathcal{V}) \in \mathcal{V}$, so the second term $(\mathbb{E}(X|\mathcal{V}) - Y) \in \mathcal{V}$. Using the property of conditional expectation, we have $\mathbb{E}(\mathbb{E}(X|\mathcal{V}) - Y|\mathcal{V}) = \mathbb{E}(X|\mathcal{V}) - Y$.

Now, let's calculate the conditional expectation of the product of $(X - \mathbb{E}(X|\mathcal{V}))$ and $(\mathbb{E}(X|\mathcal{V}) - Y)$:

$$\mathbb{E}((X - \mathbb{E}(X|\mathcal{V}))(\mathbb{E}(X|\mathcal{V}) - Y)|\mathcal{V}) = \mathbb{E}((X - \mathbb{E}(X|\mathcal{V}))(\mathbb{E}(X|\mathcal{V}) - Y)) = 0$$

The last equality follows from the fact that the product of the two terms is uncorrelated, and their expectation is zero. This implies that $(X - \mathbb{E}(X|\mathcal{V}))$ and $(\mathbb{E}(X|\mathcal{V}) - Y)$ are orthogonal.

Now, we can use the Pythagorean theorem for Hilbert spaces:

$$|X - Y|^2 = |X - \mathbb{E}(X|\mathcal{V})|^2 + |\mathbb{E}(X|\mathcal{V}) - Y|^2$$

Notice that the right-hand side is minimized when $|\mathbb{E}(X|\mathcal{V}) - Y|^2$ is minimized. This is true because $|X - \mathbb{E}(X|\mathcal{V})|^2$ is constant and non-negative. Therefore, the least square problem is minimized when $Y = \mathbb{E}(X|\mathcal{V})$, i.e., $Z = \mathbb{E}(X|\mathcal{V})$.

Hence, we have proved that $Z = \mathbb{E}(X|\mathcal{V})$ solves the least square problem:

$$|Z - X| = \min_{Y \in \mathcal{V}} |Y - X|.$$

## B.2 Relationships Among Three Diffusion Losses

In this section, we give the forms of three commonly used losses, $\epsilon$-loss, $s$-loss and $x_0$-loss in the PSM setting. Assume $\alpha = 1$ and $k_B T = 1$ for simplicity, since

$$
\begin{aligned}
\nabla_{\boldsymbol{x}_t} \log p(\boldsymbol{x}_t) &= \frac{\nabla_{\boldsymbol{x}_t} \int p(\boldsymbol{x}_0) p(\boldsymbol{x}_t|\boldsymbol{x}_0) d\boldsymbol{x}_0}{p(\boldsymbol{x}_t)} \\
&= \frac{\int p(\boldsymbol{x}_0) \left( -\nabla_{\boldsymbol{x}_0} e^{-\frac{\|\boldsymbol{x}_t - \alpha_t \boldsymbol{x}_0\|^2}{2\sigma_t^2}} \right) d\boldsymbol{x}_0}{p(\boldsymbol{x}_t)} = \int p(\boldsymbol{x}_0|\boldsymbol{x}_t) \nabla_{\boldsymbol{x}_0} \log p(\boldsymbol{x}_0) d\boldsymbol{x}_0 = \mathbb{E}_{\boldsymbol{x}_0|\boldsymbol{x}_t} \left[ \frac{-\nabla_{\boldsymbol{x}_0} \mathcal{E}}{k_B T} \right] \quad \text{(a)} \\
&= \frac{\int p(\boldsymbol{x}_0)(-\frac{\boldsymbol{x}_t - \boldsymbol{x}_0}{\sigma_t^2}) p(\boldsymbol{x}_t|\boldsymbol{x}_0) d\boldsymbol{x}_0}{p(\boldsymbol{x}_t)} = \int p(\boldsymbol{x}_0|\boldsymbol{x}_t)(-\frac{\boldsymbol{x}_t - \boldsymbol{x}_0}{\sigma_t^2}) d\boldsymbol{x}_0 = \mathbb{E}_{\boldsymbol{x}_0|\boldsymbol{x}_t} \left[ \frac{\boldsymbol{x}_0 - \boldsymbol{x}_t}{\sigma_t^2} \right] \quad \text{(b)},
\end{aligned}
\tag{25}
$$

where $\mathcal{E}$ is the energy of $x_0$. When using a score network $s_\theta$ to learn the gradient of log probability $\nabla_{\boldsymbol{x}_t} \log p(\boldsymbol{x}_t)$, the $s$-loss can be derived by (a) that

$$\mathcal{L}_{\text{score - model}} = \mathbb{E}_{t \sim U(0,1)} \mathbb{E}_{\boldsymbol{x}_t \sim p(\boldsymbol{x}_t|\boldsymbol{x}_0)} \mathbb{E}_{\boldsymbol{x}_0} \|s_\theta(\boldsymbol{x}_t, t) - (-\nabla_{\boldsymbol{x}_0} \mathcal{E})\|^2. \tag{26}$$

and accordingly, the $\epsilon$-loss is

$$\mathcal{L}_{\epsilon \text{ - model}} = \mathbb{E}_{t \sim U(0,1)} \mathbb{E}_{\boldsymbol{x}_t \sim p(\boldsymbol{x}_t | \boldsymbol{x}_0)} \mathbb{E}_{\boldsymbol{x}_0} \| \epsilon_\theta(\boldsymbol{x}_t, t) - \sigma_t \nabla_{\boldsymbol{x}_0} \mathcal{E} \|^2 , \tag{27}$$

if we take $\sigma_t$, which is independent of $x_t$ and $\theta$, out of the norm. Similarly by (b), the formula $\mathbb{E}_{\boldsymbol{x}_0 | \boldsymbol{x}_t} \left[ \frac{\boldsymbol{x}_0 - \boldsymbol{x}_t}{\sigma_t^2} + \nabla_{\boldsymbol{x}_0} \mathcal{E}(\boldsymbol{x}_0) \right]$ is expected to be zero, as the optimal score function satisfies that $\nabla_{\boldsymbol{x}_t} \log p(\boldsymbol{x}_t) = \mathbb{E}_{\boldsymbol{x}_0 | \boldsymbol{x}_t} [-\nabla_{\boldsymbol{x}_0} \mathcal{E}(\boldsymbol{x}_0)]$. Since $\mathbb{E}_{\boldsymbol{x}_0 | \boldsymbol{x}_t} \left[ \frac{\boldsymbol{x}_0 - \boldsymbol{x}_t}{\sigma_t^2} + \nabla_{\boldsymbol{x}_0} \mathcal{E}(\boldsymbol{x}_0) \right] = \mathbb{E}_{\boldsymbol{x}_0 | \boldsymbol{x}_t} \left[ \frac{\boldsymbol{x}_0 - \boldsymbol{x}_t + \sigma_t^2 \nabla_{\boldsymbol{x}_0} \mathcal{E}(\boldsymbol{x}_0)}{\sigma_t^2} \right]$, we can use the loss function with a network representing $\boldsymbol{x}_0$, $\mathcal{D}_\theta$, to learn the diffusion.

$$\mathcal{L}_{\boldsymbol{x}_0 - \text{model}} = \mathbb{E}_{t \sim U(0,1)} \mathbb{E}_{\boldsymbol{x}_t \sim p(\boldsymbol{x}_t | \boldsymbol{x}_0)} \mathbb{E}_{\boldsymbol{x}_0} \| \mathcal{D}_\theta - \boldsymbol{x}_t - \sigma_t^2 \nabla_{\boldsymbol{x}_0} \mathcal{E}(\boldsymbol{x}_0) \|_2^2 . \tag{28}$$

Therefore, $\epsilon$-loss and $x_0$-loss are different objects, but are essentially derived from the same formula.

### B.3 Proof of Theorem 2

*Proof.* Let $I_1$ be the formula on the left side $\mathbb{E}_{q(\boldsymbol{x}_0 | \boldsymbol{x}_t)} [\nabla \log p(\boldsymbol{x}_0)] - \mathbb{E}_{p(\boldsymbol{x}_0 | \boldsymbol{x}_t)} [\nabla \log p(\boldsymbol{x}_0)]$, and $I_2$ be the right one, $\mathbb{E}_{q(\boldsymbol{x}_0 | \boldsymbol{x}_t)} [\nabla \log q(\boldsymbol{x}_0)] - \mathbb{E}_{p(\boldsymbol{x}_0 | \boldsymbol{x}_t)} [\nabla \log p(\boldsymbol{x}_0)]$.

$$\begin{aligned} I_1 &= \int \nabla \log p(\boldsymbol{x}_0) \left( q(\boldsymbol{x}_0 \mid \boldsymbol{x}_t) - p(\boldsymbol{x}_0 \mid \boldsymbol{x}_t) \right) d\boldsymbol{x}_0 , \\ I_2 &= \int \left( \nabla \log q(\boldsymbol{x}_0) q(\boldsymbol{x}_0 \mid \boldsymbol{x}_t) - \nabla \log p(\boldsymbol{x}_0) p(\boldsymbol{x}_0 \mid \boldsymbol{x}_t) \right) d\boldsymbol{x}_0 , \\ I_3 &= \int \left( \nabla \log q(\boldsymbol{x}_0) - \nabla \log p(\boldsymbol{x}_0) \right) q(\boldsymbol{x}_0 \mid \boldsymbol{x}_t) d\boldsymbol{x}_0 . \end{aligned} \tag{29}$$

Then $I_2 = I_1 + I_3$. We only need to prove that for $t$ near 0,

$$\begin{aligned} \|I_1\|_2^2 &= \left\| \int \nabla \log p(\boldsymbol{x}_0) \left( q(\boldsymbol{x}_0 \mid \boldsymbol{x}_t) - p(\boldsymbol{x}_0 \mid \boldsymbol{x}_t) \right) d\boldsymbol{x}_0 \right\|_2^2 \\ &\leqslant \left\| \int \left( \nabla \log q(\boldsymbol{x}_0) q(\boldsymbol{x}_0 \mid \boldsymbol{x}_t) - \nabla \log p(\boldsymbol{x}_0) p(\boldsymbol{x}_0 \mid \boldsymbol{x}_t) \right) d\boldsymbol{x}_0 \right\|_2^2 = \|I_2\|_2^2 . \end{aligned} \tag{30}$$

By $\|I_2\|_2^2 = \|I_3 + I_1\|_2^2 = \|I_3\|_2^2 + \|I_1\|_2^2 + 2I_1^\top \cdot I_3$, we need to explain that $I_3^\top (I_3 + 2I_1) \geqslant 0$.

For simplicity, we assume the reverse SDE satisfies that $d\boldsymbol{x}_t = -h(t) \nabla_{\boldsymbol{x}_t} \log p(\boldsymbol{x}_t) dt + \sqrt{2h(t)} d\widetilde{W}_t$. Here $h(t) = \sigma_{\min}^2 (\frac{\sigma_{\max}}{\sigma_{\min}})^{2t}$, and $\widetilde{W}_t$ is a Brownian motion. For function $f, f : [0, 1] \times \mathbb{R}^d \to \mathbb{R}$, the Ito formula of $f(\boldsymbol{x}_t)$ is expanded as:

$$\begin{aligned} df(\boldsymbol{x}_t) &= f'(\boldsymbol{x}_t) d\boldsymbol{x}_t + \frac{1}{2} f''(\boldsymbol{x}_t) (d\boldsymbol{x}_t)^2 \\ &= f'(\boldsymbol{x}_t) \left[ -h(t) \nabla_{\boldsymbol{x}_t} \log p(\boldsymbol{x}_t) dt + \sqrt{2h(t)} d\tilde{W}_t \right] + \frac{1}{2} f''(\boldsymbol{x}_t) \cdot 2h(t) dt \\ &= \left[ -f'(\boldsymbol{x}_t) h(t) \nabla_{\boldsymbol{x}_t} \log p(\boldsymbol{x}_t) + f''(\boldsymbol{x}_t) h(t) \right] dt + \sqrt{2h(t)} f'(\boldsymbol{x}_t) d\widetilde{W}_t \end{aligned}$$

For reverse transition $p(\boldsymbol{x}_t \mid \boldsymbol{x}_s)$, where $t < s$, assume that $p(\boldsymbol{x}_t \mid \boldsymbol{x}_s)$ in the form of $N(g(\boldsymbol{x}_s), \beta_{t,s}^2)$, $g(\boldsymbol{x}_s)$ is a function of $\boldsymbol{x}_s$, and $\beta_{t,s}$ is a time function related to $t, s$. Then

$$\begin{aligned} dp(\boldsymbol{x}_t \mid \boldsymbol{x}_s) = &\left[ -\frac{\boldsymbol{x}_t - g(\boldsymbol{x}_s)}{\beta_{t,s}^2} p(\boldsymbol{x}_t \mid \boldsymbol{x}_s) h(t) \nabla_{\boldsymbol{x}_t} \log p(\boldsymbol{x}_t) \right. \\ &\left. + \left( -\frac{1}{\beta_{t,s}^2} p(\boldsymbol{x}_t \mid \boldsymbol{x}_s) + \left( \frac{\boldsymbol{x}_t - g(\boldsymbol{x}_s)}{\beta_{t,s}^2} \right)^2 p(\boldsymbol{x}_t \mid \boldsymbol{x}_s) \right) h(t) \right] dt + \left( -\frac{\boldsymbol{x}_t - g(\boldsymbol{x}_s)}{\beta_{t,s}^2} \sqrt{2h(t)} d\tilde{W}_t \right) . \end{aligned}$$

Therefore, $\frac{\partial p(\boldsymbol{x}_t|\boldsymbol{x}_s)}{\partial t} = p\left(\boldsymbol{x}_t \mid \boldsymbol{x}_s\right) h(t) \left[-\frac{\boldsymbol{x}_t - g(\boldsymbol{x}_s)}{\beta_{t,s}^2}\nabla_{\boldsymbol{x}_t}\log p\left(\boldsymbol{x}_t\right) - \frac{1}{\beta_{t,s}^2} + \left(\frac{\boldsymbol{x}_t - g(\boldsymbol{x}_s)}{\beta_{t,s}^2}\right)^2\right]$, and

$$p\left(\boldsymbol{x}_0 \mid \boldsymbol{x}_s\right) = p\left(\boldsymbol{x}_s \mid \boldsymbol{x}_s\right) - \left[p\left(\boldsymbol{x}_0 \mid \boldsymbol{x}_s\right) - h(0)\left(-\frac{\boldsymbol{x}_0 - g\left(\boldsymbol{x}_s\right)}{\beta_{0,s}^2}\nabla_{\boldsymbol{x}_0}\log p\left(\boldsymbol{x}_0\right) - \frac{1}{\beta_{0,s}^2} + \left(\frac{\boldsymbol{x}_0 - g\left(\boldsymbol{x}_s\right)}{\beta_{0,s}^2}\right)^2\right)\right] \cdot s + O\left(s^2\right)$$

$$= p\left(\boldsymbol{x}_s\right) - \left[h(0)p\left(\boldsymbol{x}_0 \mid \boldsymbol{x}_s\right)\frac{1}{\beta_{0,s}^2}\left[\left(g\left(\boldsymbol{x}_s\right) - \boldsymbol{x}_0\right)\nabla_{\boldsymbol{x}_0}\log p\left(\boldsymbol{x}_0\right) - 1 + \left(\frac{\left(\boldsymbol{x}_0 - g\left(\boldsymbol{x}_s\right)\right)}{\beta_{0,s}}\right)^2\right]\right] \cdot s + O(s^2).$$

Let $C_1\left(\boldsymbol{x}_0, \boldsymbol{x}_s\right) = -h(0)p\left(\boldsymbol{x}_0 \mid \boldsymbol{x}_s\right)\frac{1}{\beta_{0,s}^2}\cdot\left(\left(g\left(\boldsymbol{x}_s\right) - \boldsymbol{x}_0\right)\nabla_{\boldsymbol{x}_0}\log p\left(\boldsymbol{x}_0\right) - 1 + \left(\frac{\boldsymbol{x}_0 - g(\boldsymbol{x}_s)}{\beta_{0,s}}\right)^2\right)$, then $p\left(\boldsymbol{x}_0|\boldsymbol{x}_s\right) = p\left(\boldsymbol{x}_s\right) + C_1\left(\boldsymbol{x}_0, \boldsymbol{x}_s\right)\cdot s + O\left(s^2\right)$.

$$I_3 = \int\left(\nabla\log q\left(\boldsymbol{x}_0\right) - \nabla\log p\left(\boldsymbol{x}_0\right)\right)\left[p\left(\boldsymbol{x}_t\right) + C_1\left(\boldsymbol{x}_0, \boldsymbol{x}_t\right)t + O\left(t^2\right)\right]d\boldsymbol{x}_0$$

$$= p\left(\boldsymbol{x}_t\right)\int\left(\nabla\log q\left(\boldsymbol{x}_0\right) - \nabla\log p\left(\boldsymbol{x}_0\right)\right)d\boldsymbol{x}_0 + t\cdot\int\left(\nabla\log q\left(\boldsymbol{x}_0\right) - \nabla\log p\left(\boldsymbol{x}_0\right)\right)\cdot C_1\left(\boldsymbol{x}_0, \boldsymbol{x}_t\right)d\boldsymbol{x}_0 + O\left(t^2\right),$$

where $C_2\left(\boldsymbol{x}_0, \boldsymbol{x}_s\right) = -h(0)q\left(\boldsymbol{x}_0 \mid \boldsymbol{x}_s\right)\frac{1}{\beta_{0,s}^2}\cdot\left(\left(g\left(\boldsymbol{x}_s\right) - \boldsymbol{x}_0\right)\nabla_{\boldsymbol{x}_0}\log q\left(\boldsymbol{x}_0\right) - 1 + \left(\frac{\boldsymbol{x}_0 - g(\boldsymbol{x}_s)}{\beta_{0,s}}\right)^2\right)$, then $p\left(\boldsymbol{x}_0|\boldsymbol{x}_s\right) = p\left(\boldsymbol{x}_s\right) + C_1\left(\boldsymbol{x}_0, \boldsymbol{x}_s\right)\cdot s + O\left(s^2\right)$.

We then expand $I_1 := \int\nabla_{\boldsymbol{x}_0}\log p\left(\boldsymbol{x}_0\right)\left(q\left(\boldsymbol{x}_0 \mid \boldsymbol{x}_t\right) - p\left(\boldsymbol{x}_0 \mid \boldsymbol{x}_t\right)\right)d\boldsymbol{x}_0$. From above, we know that,

$$I_1 = \int\nabla\log p\left(\boldsymbol{x}_0\right)\left[q\left(\boldsymbol{x}_t\right) + C_2\left(\boldsymbol{x}_0, \boldsymbol{x}_t\right)t - p\left(\boldsymbol{x}_t\right) - C_1\left(\boldsymbol{x}_0, \boldsymbol{x}_t\right)t + O\left(t^2\right)\right]$$

$$= \left(q\left(\boldsymbol{x}_t\right) - p\left(\boldsymbol{x}_t\right)\right)\int\nabla\log p\left(\boldsymbol{x}_0\right)d\boldsymbol{x}_0 + \int\nabla\log p\left(\boldsymbol{x}_0\right)\left(C_2\left(\boldsymbol{x}_0, \boldsymbol{x}_t\right) - C_1\left(\boldsymbol{x}_0\right)\boldsymbol{x}_t\right)d\boldsymbol{x}_0 + O\left(t^2\right).$$

$$2I_3^\top I_1 = 2p\left(\boldsymbol{x}_t\right)\cdot\left(q\left(\boldsymbol{x}_t\right) - p\left(\boldsymbol{x}_t\right)\right)\left(\int\left(\nabla\log q\left(\boldsymbol{x}_0\right) - \nabla\log p\left(\boldsymbol{x}_0\right)\right)d\boldsymbol{x}_0\right)^\top\left(\int\nabla\log p\left(\boldsymbol{x}_0\right)d\boldsymbol{x}_0\right)$$

$$+ 2t\left(q\left(\boldsymbol{x}_t\right) - p\left(\boldsymbol{x}_t\right)\right)\left(\int C_1\left(\boldsymbol{x}_0, \boldsymbol{x}_t\right)\left(\nabla\log q\left(\boldsymbol{x}_0\right) - \nabla\log p\left(\boldsymbol{x}_0\right)\right)d\boldsymbol{x}_0\right)^\top\cdot\left(\int\nabla\log p\left(\boldsymbol{x}_0\right)d\boldsymbol{x}_0\right)$$

$$+ 2t\,p\left(\boldsymbol{x}_t\right)\left(\int\left(C_2\left(\boldsymbol{x}_0, \boldsymbol{x}_t\right) - C_1\left(\boldsymbol{x}_0, \boldsymbol{x}_t\right)\right)\nabla\log p\left(\boldsymbol{x}_0\right)d\boldsymbol{x}_0\right)^\top\cdot\left(\int\left(\nabla\log q\left(\boldsymbol{x}_0\right) - \nabla\log p\left(\boldsymbol{x}_0\right)\right)d\boldsymbol{x}_0\right)$$

$$+ 2t^2\cdot\left(\int\left(\nabla\log q\left(\boldsymbol{x}_0\right) - \nabla\log p\left(\boldsymbol{x}_0\right)\right)\cdot C_1\left(\boldsymbol{x}_0, \boldsymbol{x}_t\right)d\boldsymbol{x}_0\right)^\top\cdot\left(\int\nabla\log p\left(\boldsymbol{x}_0\right)\left(C_2\left(\boldsymbol{x}_0, \boldsymbol{x}_t\right) - C_1\left(\boldsymbol{x}_0, \boldsymbol{x}_t\right)\right)d\boldsymbol{x}_0\right)$$

$$+ O\left(t^3\right)$$

Since $\int\nabla\log p\left(\boldsymbol{x}_0\right)d\boldsymbol{x}_0 = 0$,

$$2I_3^\top I_1 = 2t^2\cdot\left(\int\left(\nabla\log q\left(\boldsymbol{x}_0\right) - \nabla\log p\left(\boldsymbol{x}_0\right)\right)\cdot C_1\left(\boldsymbol{x}_0, \boldsymbol{x}_t\right)d\boldsymbol{x}_0\right)^\top\cdot\left(\int\nabla\log p\left(\boldsymbol{x}_0\right)\left(C_2\left(\boldsymbol{x}_0, \boldsymbol{x}_t\right) - C_1\left(\boldsymbol{x}_0, \boldsymbol{x}_t\right)\right)d\boldsymbol{x}_0\right)$$

$$+ O\left(t^3\right)$$

Consider the second term of the last equality,

$$C_2\left(\boldsymbol{x}_0, \boldsymbol{x}_t\right) - C_1\left(\boldsymbol{x}_0, \boldsymbol{x}_t\right) = h(0)\frac{1}{\beta_{0,s}^2}\left[\left(g\left(\boldsymbol{x}_s\right) - \boldsymbol{x}_0\right)\left(p\left(\boldsymbol{x}_0 \mid \boldsymbol{x}_s\right)\nabla_{\boldsymbol{x}_0}\log p\left(\boldsymbol{x}_0\right) - q\left(\boldsymbol{x}_0 \mid \boldsymbol{x}_s\right)\nabla_{\boldsymbol{x}_0}\log q\left(\boldsymbol{x}_0\right)\right)\right]$$

$$+ h(0)\frac{1}{\beta_{0,s}^2}\left[\left(-1 + \left(\frac{\boldsymbol{x}_0 - g\left(\boldsymbol{x}_s\right)}{\beta_{0,s}}\right)^2\right)\left(p\left(\boldsymbol{x}_0 \mid \boldsymbol{x}_s\right) - q\left(\boldsymbol{x}_0 \mid \boldsymbol{x}_s\right)\right)\right]$$

Then

$$\int \nabla \log p\left(\boldsymbol{x}_0\right) - \left(C_2\left(\boldsymbol{x}_0, \boldsymbol{x}_t\right) - C_1\left(\boldsymbol{x}_0, \boldsymbol{x}_t\right)\right) d\boldsymbol{x}_0 = h(0) \cdot \int \left(\frac{\boldsymbol{x}_0 - g\left(\boldsymbol{x}_t\right)}{\beta_{0,t}^2}\right)^2 \cdot \left(p\left(\boldsymbol{x}_0 \mid \boldsymbol{x}_t\right) - q\left(\boldsymbol{x}_0 | \boldsymbol{x}_t\right)\right) \cdot \nabla \log p\left(\boldsymbol{x}_0\right) d\boldsymbol{x}_0$$

$$= h(0) \cdot \int - \left(\frac{\boldsymbol{x}_0 - g\left(\boldsymbol{x}_t\right)}{\beta_{0,t}^2}\right)^2 \cdot p\left(\boldsymbol{x}_t | \boldsymbol{x}_0\right) \left(\frac{p\left(\boldsymbol{x}_0\right)}{p\left(\boldsymbol{x}_t\right)} - \frac{q\left(\boldsymbol{x}_0\right)}{q\left(\boldsymbol{x}_t\right)}\right) \nabla \log p\left(\boldsymbol{x}_0\right) d\boldsymbol{x}_0$$

$$\underset{(i)}{=} O(t) \cdot h(0) \int \left(\frac{\boldsymbol{x}_0 - g\left(\boldsymbol{x}_t\right)}{\beta_{0,t}^2}\right)^2 p\left(\boldsymbol{x}_t \mid \boldsymbol{x}_0\right) \nabla \log p\left(\boldsymbol{x}_0\right) d\boldsymbol{x}_0$$

The $(i)$ above is the assumption $\frac{p(\boldsymbol{x}_0)}{p(\boldsymbol{x}_t)} \approx 1$, which is result from that for $p(\boldsymbol{x}_t | \boldsymbol{x}_0) = \mathcal{N}(\boldsymbol{x}_0, \sigma_t^2 I)$, where $\sigma_t = \sigma_{\min} \left(\frac{\sigma_{\max}}{\sigma_{\min}}\right)^t$, and expanding $p(\boldsymbol{x}_t)$ around $\boldsymbol{x}_0$ is $p(\boldsymbol{x}_t) = p(\boldsymbol{x}_0) + \frac{1}{2}\sigma_0^2 \nabla^2 p(\boldsymbol{x}_0) t + O(t^2)$. Hence, $\frac{p(\boldsymbol{x}_0)}{p(\boldsymbol{x}_t)} = O(t)$. Then

$$2I_3^\top I_1 = 2t^3 \left(\int C_1\left(\boldsymbol{x}_0, \boldsymbol{x}_t\right) \left(\nabla \log q - \nabla \log p\right) d\boldsymbol{x}_0\right)^\top \cdot \left(h(0) \int \left(\frac{\boldsymbol{x}_0 - g(\boldsymbol{x}_t)}{\beta_{0,t}^2}\right)^2 p\left(\boldsymbol{x}_t \mid \boldsymbol{x}_0\right) \nabla \log p\left(\boldsymbol{x}_0\right) d\boldsymbol{x}_0\right).$$

Therefore,

$$I_3^\top \left(I_3 + 2I_1\right) = I_3^T I_3 + O(t^3)$$

$$= \left(\int \left(\nabla \log q\left(\boldsymbol{x}_0\right) - \nabla \log p\left(\boldsymbol{x}_0\right)\right) q\left(\boldsymbol{x}_0 \mid \boldsymbol{x}_t\right) d\boldsymbol{x}_0\right)^T \left(\int \left(\nabla \log q\left(\boldsymbol{x}_0\right) - \nabla \log p\left(\boldsymbol{x}_0\right)\right) q\left(\boldsymbol{x}_0 \mid \boldsymbol{x}_t\right) d\boldsymbol{x}_0\right) + O(t^3).$$

If $t$ is small, $q(\boldsymbol{x}_0 | \boldsymbol{x}_t) \approx \delta(\boldsymbol{x}_0 - f(\boldsymbol{x}_t))$, where $f$ is a function of $\boldsymbol{x}_t$. Since $\nabla \log q\left(\boldsymbol{x}_0\right) \neq \nabla \log p\left(\boldsymbol{x}_0\right)$, then $I_3^\top \left(I_3 + 2I_1\right) > 0$ when $t$ is small. $\qquad \square$

## B.4 Singularity and Variance Problem

This section explains the role of PSM in the neighborhood of $t = 0$ from two aspects: (1) PSM has a smaller training variance near 0 and the training is more stable; (2) PSM can alleviate the problem that the Lipschitz constant of DSM is too large at 0.

**(1) Variance of PSM.** Let $X$ be a sample from the data distribution, and let $Y$ be a sample from the noisy process $\boldsymbol{x}_t = \alpha_t \boldsymbol{x}_0 + \sigma_t \boldsymbol{\epsilon}$.

**Lemma 2.** *When $t$ is close to zero, the variance of the DSM target satisfies $Var_{X|Y}\left(\nabla \log p(Y|X)\right) \gg 0$ if the noise schedule close to 0 near $t = 0$, and the variance of the PSM target $Var_{X|Y}(\nabla \log p(X))$ is smaller than DSM when $t$ near 0.*

*Proof.* Let $X$ is the sample from the data distribution and $Y$ is the noised data, $Y = \alpha X + \sigma_t \varepsilon$, we can prove the variance of PSM target $Var_{X|Y}(\nabla \log p(X))$ is smaller than the DSM target $Var_{X|Y}(\nabla \log p(y|x))$ near $t = 0$ while larger when $t$ is larger. The proof strategy here is similar to that of TSM Bortoli et al. (2024); however, their work only considers the Gaussian case, whereas we address more complex distributions. Nevertheless, the corresponding Boltzmann distribution or data distribution can be approximated by mixture Gaussian distribution, and thus it suffices to establish the desired properties under the Gaussian setting for simplicity. Let $d$ be the dimension of the problem. Assume that $p_X(x) = \mathcal{N}\left(\mu, \sigma_{\text{tar}}^2\right)$, and the conditional probability

$$p_{Y|X}(y \mid x) = \mathcal{N}\left(\alpha_t x, \sigma_t^2\right). \tag{31}$$

By convolution, $p_Y(y) = \mathcal{N}\left(\alpha_t \mathbb{E}(X), \alpha_t^2 \sigma_{\text{tar}}^2 + \sigma_t^2\right)$. Then, by Bayes' formula, $p_{X|Y}(x \mid y) = \frac{P_X(x) \cdot p_{Y|X}(y)}{P_Y(y)} \propto \frac{\mathcal{N}\left(\mu^{(1)}, \sigma^{(1)^2}\right)}{P_Y(y)}$.

$$p_X(x) \cdot p_{Y|X}(y \mid x) \propto \exp\left\{-\frac{(x - \mu)^2}{2\sigma_{\text{tar}}^2} - \frac{(y - \alpha_t x)^2}{2\sigma_t^2}\right\}$$

$$= \exp\left\{\frac{1}{2\sigma_t^2\sigma_{\text{tar}}^2}\left[\sigma_t^2\left(x^2+\mu^2-\mu x\right)+\sigma_{\text{tar}}^2\left(y^2+\alpha_t^2 x^2-2\alpha_t xy\right)\right]\right\}$$

$$= \exp\left\{\frac{1}{2\sigma_t^2\sigma_{\text{tar}}^2}\left[\left(\alpha_t^2\sigma_{\text{tar}}^2+\sigma_t^2\right)x^2-2\left(\mu\sigma_t^2+\alpha_t y\sigma_{\text{tar}}^2\right)\boldsymbol{x}+\mu^2\sigma_t^2+y^2\sigma_{tar}^2\right]\right\}.$$

Hence, $\sigma^{(1)^2}=\frac{\sigma_t^2\sigma_{\text{tar}}^2}{\alpha_t^2\sigma_{\text{tar}}^2+\sigma_t^2}$, and

$$p_{X|Y}(x\mid y)=\frac{\mathcal{N}\left(\mu^{(1)},\sigma^{(1))^2}\right)}{P_Y(y)}=\exp\left\{-\frac{\left(x-\mu^{(1)}\right)^2}{2\sigma^{(1)2}}+\frac{(y-\alpha_t\mathbb{E}(x))^2}{2\left(\sigma_t^2+\alpha_t\sigma_{tar}^2\right)}\right\}$$

$$=\exp\left\{-\frac{1}{2(\sigma^{(1)})^2}\left[\left(x-\mu^{(1)}\right)^2-\frac{\sigma^{(1))^2}}{\sigma_t^2+\alpha_t\sigma_{\text{tar}}^2}(y-\alpha_t\mathbb{E}(x))^2\right]\right\}$$

$$=\exp\left\{-\frac{1}{2(\sigma^{(1)})^2}\left[\left(x-C\left(\mu^{(1)},\sigma^{(1)},\mu,\alpha_t,\sigma_{\text{tar}}^2,\mathbb{E}(x)\right)\right)^2\right]\right\},$$

where $C(\cdot)$ is a constant decided by $\cdot$. Therefore, $\text{Var}_{X|Y}(x)=\sigma^{(1)^2}=\frac{\sigma_t^2\sigma_{\text{tar}}^2}{\alpha_t^2\sigma_{\text{tar}}^2+\sigma_t^2}$. Therefore,

$$\sum_{i=1}^{d}\text{Var}_{X|Y}\left(\nabla_i\log p_{Y|X}(y\mid X=x)\right)=\sum_{i=1}^{d}\text{Var}_{X|Y}\left(\frac{\alpha_t X-y}{\sigma_t{}^2}\right)=\sum_{i=1}^{d}\frac{\alpha_t^2}{\sigma_t^4}\text{Var}_{X|Y}(X)$$

$$=\sum_{i=1}^{d}\alpha_t\frac{\alpha_t^2}{\sigma_t^2}\frac{\sigma_{\text{tar}}^2}{\alpha_t^2\sigma_{\text{tar}}^2+\sigma_t^2}=d\alpha_t^2\left(\frac{\sigma_{\text{tar}}}{\sigma_t}\right)^2\frac{1}{\alpha_t^2\sigma_{\text{tar}}^2+\sigma_t^2}, \tag{32}$$

and

$$\sum_{i=1}^{d}\text{Var}_{X|Y}\left(\nabla_i\log p(x)\right)=\sum_{i=1}^{d}\text{Var}_{X|Y}\left(\frac{x-\mu}{\sigma_{tar}^2}\right)=\frac{d}{\alpha_t^2}\left(\frac{\sigma_t}{\sigma_{\text{tar}}}\right)^2\frac{1}{\alpha_t^2\sigma_{\text{tar}}^2+\sigma_t^2}. \tag{33}$$

$\square$

**(2) The Lipschitz problem of DSM.**

**Lemma 3.** *Given the forward process $\boldsymbol{x}_t=\alpha_t\boldsymbol{x}_0+\sigma_t\boldsymbol{\epsilon}$, if the noise schedule satisfies $\sigma_t\to 0$ as $t\to 0$, then the Lipschitz constant of the neural network $\epsilon_\theta$ satisfies $\lim_{t\to 0}\sup\left\|\frac{\partial\epsilon_\theta(\boldsymbol{x}_t,t)}{\partial t}\right\|_2\to\infty$.*

*Proof.* By $\epsilon_\theta=\sigma_t\cdot s_\theta\approx\sigma_t\cdot\nabla_x\log p(\boldsymbol{x}_t)$ and the chain rule, $\frac{\partial\epsilon_\theta}{\partial t}=-\frac{d\sigma_t}{dt}\nabla_x\log p(\boldsymbol{x}_t)-\frac{\partial\nabla_x\log p(\boldsymbol{x}_t)}{\partial t}\sigma_t$. We assume that the $\nabla_x\log p(\boldsymbol{x}_t)$ is smooth. Since $\boldsymbol{x}_t=\alpha_t\boldsymbol{x}_0+\sigma_t\boldsymbol{\epsilon}$,

(i) VESDE: $\alpha_t=1$, $\sigma_t=\sigma_0\left(\frac{\sigma_1}{\sigma_0}\right)^t$, where $\sigma_0=\sigma_{\min}$, $\sigma_1=\sigma_{\max}$, then $\frac{d\sigma_t}{dt}=\sigma_0\left(\frac{\sigma_1}{\sigma_0}\right)^t\cdot\ln\frac{\sigma_1}{\sigma_0}=\sigma_t\cdot\ln\frac{\sigma_1}{\sigma_0}$. Therefore, $\frac{d\sigma_t}{dt}\nabla_x\log p(\boldsymbol{x}_t)$ has the order of $O(\ln\frac{\sigma_1}{\sigma_0}\cdot\boldsymbol{z})$, where $\boldsymbol{z}$ is the random noise. As Song et al. (2020b) mentioned that, $\sigma_0$ approaches 0 theoretically, but in practice, $\sigma_0=0.01$ is taken to avoid singular values.

(ii) VPSDE: $\alpha_{t=}e^{-\frac{1}{2}\int_0^t\beta_s d_s}$, $\sigma_t=\sqrt{1-e^{-\int_0^t\beta_s dt}}$, then $\frac{d\sigma_t}{dt}=-\beta_t\left(-e^{-\int_0^t\beta_s ds}\right)\left(1-e^{-\int_0^t\beta_s ds}\right)^{-\frac{1}{2}}=$

$\frac{\beta_t e^{-\int_0^t\beta_s ds}}{\sqrt{1-e^{-\int_0^t\beta_s ds}}}\to+\infty$    if $t\to 0^+$.

It shows the Lipschitz singularity property. $\square$

In summary, our theoretical analysis (Lemma 2, Lemma 3 and Theorem 2) demonstrates that PSM reduces variance and corrects estimation bias in the small-t regime, providing strong motivation for its use when $t$ is close to zero.

## C   More Experimental Details

**Dataset and settings.** The MD17 dataset and the MD22 dataset can be downloaded from `http://quantum-machine.org/gdml/data/npz`, the LJ13 and LJ55 data are from `https://osf.io/srqg7/files/osfstorage`. Hyperparameters for experiments are recorded in Table 6 and 7.

Table 6: Hyperparameters for different datasets with biased trainset

| Dataset | Data Split (train, val, test) | Batch Size (train, val) | Max Epochs | Radius of Graph |
|---|---|---|---|---|
| LJ13 | First 1k, 0.1, remaining | (64, 64) | 2000 | 4 |
| LJ55 | First 1k, 0.1, remaining | (64, 64) | 2000 | 5 |
| MD17 | First 5k, 0.1, remaining | (64, 64) | 1000 | 5 |
| MD22 (Dw nanotube) | First 5k, 0.1, remaining | (8, 8) | 3000 | 4 |
| MD22 (others) | First 5k, 0.1, remaining | (16, 16) | 500 | 7 (AT-AT) 
 7 (AT-AT-CG-CG) 
 6 (Ac-Ala3-NHMe) 
 6 (Docosahexaenoic Acid) 
 6 (Stachyose, Buckyballcatcher) |
| **Additional Settings** | | | | |

Learning rate: 0.0002    Lr scheduler: Cosine    Seed: 42

Optimizer: Adam    Weight decay: $5 \times 10^{-7}$

All diffusion parameters: $\sigma_{\min} = 0.1$, $\sigma_{\max} = 5$, time weight function $\lambda(t) = 1$

Sampling timesteps: $1,000$    Sampling method: Predict-corrector sampler

Table 7: Hyperparameters compared with previous work in Table 2.

| Dataset | Data Split (train, val, test) | Batch Size (train, val) | Max Epochs | Radius of Graph | Noise Schedule |
|---|---|---|---|---|---|
| LJ13 | 0.1, 0.1, remaining | (64, 64) | 5000 | 4 | $\sigma_{\min} = 0.01$, $\sigma_{\max} = 8$ |
| LJ55 | 0.1, 0.1, remaining | (64, 64) | 5000 | 5 | $\sigma_{\min} = 0.01$, $\sigma_{\max} = 4$ |
| Additional Settings are the same as before in Table 6. | | | | | |

**Toy models in 1D and 2D.** We conduct experiments using PSM alone in 1D and 2D toy distributions, as shown in Figure 6, where $p(x) = e^{-5\|x\|^2 + \|x\|^4}$ and the according force label is $(-4\|x\|_2^2 + 10)x$. It indicates that PSM and Piecewise perform comparably to DSM in 1D and 2D case, and it can be seen that Piecwise and PSM can better learn the sparse area in the middle. Previous work also addresses the challenges for learning high-dimensional molecular problems using only an energy label or log-density gradient Woo & Ahn (2024); Bortoli et al. (2024). The constraint of force labels $F$ become weak due to the projection of time-zero information onto highly noisy $x_t$ for large $t$ is smaller, which is unstable for dynamical learning. Refs Yang et al. (2023) and Bortoli et al. (2024) have shown that DSM suffers from high variance and singularities near $t = 0$, and encouragingly, our analysis further demonstrates that PSM achieves lower variance and corrects bias in the small-$t$ regime (Lemma 2 and Theorem 2), motivating its use specifically for small $t$. Therefore, for high dimensional molecular system, we use PSM in small $t$ range, and use DSM for large $t$.

**Lennard-Jones potential.** Here we additionally supplement the LJ potential with biased data training to obtain the energy comparisons in Figure 7.

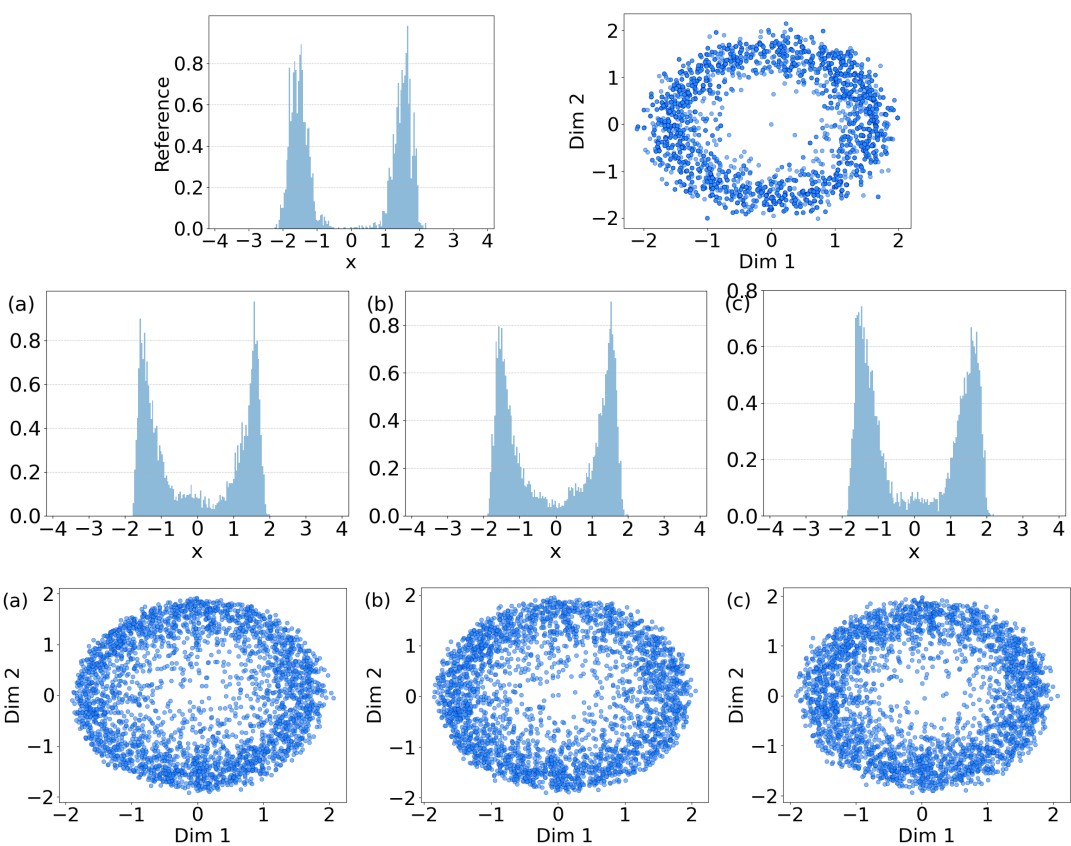

Figure 6: Comparisons of (a) DSM, (b) Piecewise, (c) PSM in one-dimensional (the second row) and in two-dimensional (the third row) examples. The first row is the ground truth for 1d and 2d cases where the exact density function is $p(x) = e^{-5\|x\|^2 + \|x\|^4}$.

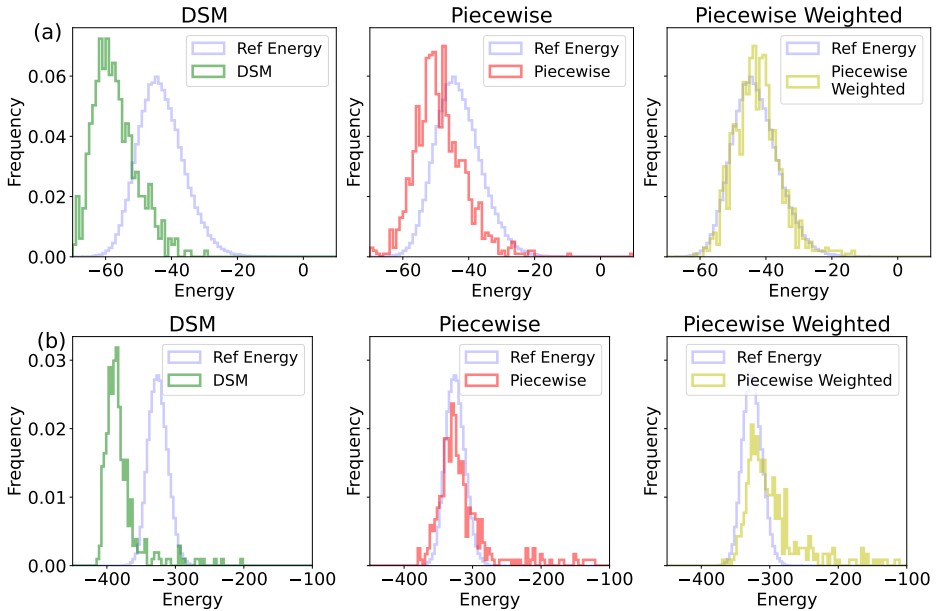

Figure 7: Comparisons of energy for (a) LJ-13 and (b) LJ-55 potential training with first $1,000$ data.

# D    Relationship with Flow Matching

Flow matching is a powerful tool for generating molecular conformations and predicting molecular properties Lipman et al. (2022); Chen & Lipman (2023); Miller et al. (2024). It enables faster training and sampling while achieving superior generalization performance. Motivated by recent advancements Woo & Ahn (2024); Domingo-Enrich et al. (2024), unified frameworks have been developed to describe both diffusion models and flow matching methods comprehensively.

Let $\boldsymbol{y}_0$ and $\boldsymbol{y}_1$ be samples from the noise distribution and the data distribution, respectively, in the flow matching framework. As demonstrated in Domingo-Enrich et al. (2024), the relationship between the velocity field and the score function is given by:

$$v(\boldsymbol{x}, t) = \frac{\dot{\alpha}_t}{\alpha_t}\boldsymbol{x} + \gamma_t \left( \frac{\dot{\alpha}_t}{\alpha_t}\gamma_t - \dot{\gamma}_t \right) s(\boldsymbol{x}, t),$$

where $\boldsymbol{x}_t = \alpha_t \boldsymbol{y}_1 + \gamma_t \boldsymbol{y}_0 = \alpha_t \boldsymbol{x}_0 + \gamma_t \varepsilon$. Based on this relationship, our method can be extended to flow matching. In our setting, the coefficients are defined as follows:

$$\text{VE}: \ \alpha_t = 1, \quad \gamma_t = \sigma_t = \sigma_0 \left( \frac{\sigma_1}{\sigma_0} \right)^t, \quad \text{VP}: \ \alpha_t = e^{-\frac{1}{2}\int_0^t \beta_s ds}, \quad \gamma_t = \left( 1 - e^{-\int_0^t \beta_s ds} \right)^{\frac{1}{2}}. \tag{34}$$

Therefore, the vector field $v(\boldsymbol{x}, t) = \gamma_t \dot{\gamma}_t s(\boldsymbol{x}, t) = \sigma_0^2 \left( \frac{\sigma_1}{\sigma_0} \right)^{2t} \ln \left( \frac{\sigma_1}{\sigma_0} \right) s(\boldsymbol{x}, t)$ for VESDE, and $v(\boldsymbol{x}, t) = -\frac{1}{2}\beta_t \boldsymbol{x} + \gamma_t \left( -\frac{1}{2}\beta_t \gamma_t - \dot{\gamma}_t \right) s(\boldsymbol{x}, t)$ for VPSDE. Building on the interplay between energy and the score function, we propose to investigate an energy-informed flow matching method. This energy-based extension will form the basis of our future research endeavors.

