# OpenReview forum: "Potential Score Matching: Debiasing Molecular Structure Sampling with Potential Energy Guidance"
_TMLR — Accepted by TMLR_

### Review · Reviewer_VW23 · 2025-04-28

**Summary Of Contributions:**

The authors present Potential Score Matching, a novel method for sampling molecular conformations from the Boltzmann distribution. A main component of PSM is a piecewise loss and a piecewise weighted loss, which prioritize PSM at small $t$ and DSM at large $t$.

The authors conduct several experiments on molecular dynamical datasets, including MD17 and MD22, which are relatively high-dimensional compared to commonly used benchmarks such as the LJ potentials.

**Audience:**

Yes

**Claims And Evidence:**

Yes

**Requested Changes:**

Please see above.

**Strengths And Weaknesses:**

**Strengths)**
- Simple modifications in the loss function, but exhibits superior performance
- Conduct experiments on several high-dimensional benchmarks.

**Weakness)**
- I have a question regarding the choice of $t_psm$ and also the weighted function $\omega_t$. While it seems that they are fixed across all experiments, it would be better to conduct some ablation studies regarding those components.

**Questions)**
- As I am not an expert on molecular dynamics. I have some questions regarding the experiments. First, for Figure 1, it seems that the suggested method more clearly approximates the reference data distribution. However, according to the captions, they are trained on the biased dataset. Could you give me more elaboration on what the figure indicates exactly?
- Similar question regarding Figure 4. It seems that all the figures look very similar. Could you give me more elaboration on what the figure indicates exactly?

---

> ### Author Response · Authors · 2025-05-26
> **Response to Reviewer VW23**
>
> We sincerely thank the reviewer for the insightful comments. Below, we provide detailed responses. The corresponding revisions have been marked in \textcolor{red}{RED} in the revised manuscript for clarity.
>
> \paragraph{Response Regarding the Choice of $t_{\text{psm}}$.}
> As established in Lemma 2 in Appendix B.3, the DSM label exhibits large training variance when $t$ is small. Furthermore, as noted in [1], DSM may encounter singularities as $t \to 0$, which can lead to unstable or inaccurate learning. This highlights the need for improved training strategies in the small-$t$ regime. Encouragingly, Theorem 2 shows that PSM offers a theoretically grounded debiasing mechanism by promoting convergence toward the Boltzmann distribution in molecular systems. This suggests the potential of applying PSM labels in the small-$t$ region of diffusion models.
>
> Motivated by this analysis, we propose a time-weighted loss function that emphasizes the contribution of PSM at small $t$ and DSM at larger $t$. We use a sigmoid-based interpolation for transition between these two regimes, i.e. the ''Piecewise-weighted loss''. Another ''Piecewise'' scheme with a hard cutoff at $t = 0.05$ can be viewed as a special case of ''Piecewise-weighted loss''. Since this general approach already demonstrates the superiority of PSM in the small-$t$ regime, we did not conduct further experiments with alternative piecewise weighting strategies.
>
>
> \paragraph{Question 1: Training on Biased Data and Approximating the Reference Distribution.}
> In molecular systems, obtaining Boltzmann distributions or accurate estimates of ensemble-averaged quantities requires sufficiently long molecular dynamics (MD) trajectories, due to the system’s ergodicity. However, generating such long trajectories is computationally expensive. Our training data consist of short MD trajectories, which represent biased samples. PSM is specifically designed to address this challenge by training on these biased, short-trajectory datasets. The results demonstrate that PSM yields distributions closer to this unbiased reference, thereby validating its ability to learn effectively from biased data and perform implicit debiasing.
>
> \paragraph{Question 2 about Figure 4.}
> Figure 4 shows the UMAP projections of SOAP descriptors, which encode high-dimensional local atomic environments rather than ensemble-averaged features such as h(r). As a nonlinear dimensionality reduction technique, UMAP preserves important structural relationships from the original feature space and can extract multiple features of the data distribution without relying on a fixed numerical threshold to indicate distributional or conformational differences. While the visual differences may appear subtle, the UMAP projections clearly divide the data into two main clusters that correspond closely to those of the reference distribution. We quantified the proportion of data points in each cluster: for PSM and DSM, the ratios are (29.4\%, 70.6\%) and (31\%, 68.4\%), respectively, compared to the reference’s (28.2\%, 71.8\%). The closer match of PSM’s cluster proportions to the reference indicates that PSM better captures local atomic environments, resulting in a debiased molecular distribution.
>
>
> Thanks again for your valuable suggestions.
>
> \paragraph{Reference:}\,\\
>
> [1] Yang, Zhantao and Feng, Ruili and Zhang, Han and Shen, Yujun and Zhu, Kai and Huang, Lianghua, et. al. Lipschitz Singularities in Diffusion Models.

---

### Review · Reviewer_whEG · 2025-05-31

**Summary Of Contributions:**

This paper proposes Potential Score Macthing (PSM) to sample from Boltzmann distribution using biased training data and the potential energy gradient. The key idea of PSM is to leverage the target score matching (TSM, Bortoli 2024) identity, which translates the score at time t to some expectation of the "clean" target score -- equivalent to the force field in the molecular simulation context. PSM can be further combined with Denoising Score Matching (DSM, the standard way of training diffusion models) to improve the empirical performance

The paper benchmarks PSM against existing methods on low-dimensional molecular dynamics (MD) systems, demonstrating competitive performance. In high-dimension MD systems, the combination of PSM and DSM outperforms using only DSM.

**Audience:**

Yes

**Claims And Evidence:**

Yes

**Requested Changes:**

For most of the requested changes, see Weakness section.

Additionally, I wonder what "energy labels" in table 1 refer to? Are they just the force (or equivalently the clean score function)? In that case, shouldn't some of the MD and MCMC methods satisfy the requirement that they don't require energy function but only the energy labels? For example, langevin dynamics only utilize the gradient information, i.e. the score function.

**Strengths And Weaknesses:**

Strengths:

- The paper is mostly nicely written and easy to follow.

- The method is straightforward an easy to implement. Compared to existing approaches like iDEM, PIS or DDS, it can be easily scaled to high dimensions. It can also easily utilize biased training data. Furthermore, theorem 2 demonstrates theoretical PSM's advantage over DSM.

- The overall empirical results seem convincing.

Weakness:

- Limited novelty: PSM is largely built upon TSM. In particular, the original TSM paper mentioned the option of combining TSM and DSM (in their page 3). And in principle, TSM can be applied to biased training data as well (contradicting the presentation in table 1). I wonder if the authors can clarify these points and highlight the novelty of the proposed method?

- Lacking theoretical guarantee: there is not theoretical guarantee that the proposed method would generate exact samples from the Boltzmann distribution. Hence i wonder if the "debiasing" claim should be reframed.

- Lacking comparison to MD: In high dimensional tasks, while it is reasonable to exclude comparison to other diffusion-based methods (as they won't easily scale), I feel it'd be nice to compare to classical MD approaches, e.g.  the number of force evaluations given the same performance, or convergence speed, or the amortization advantage.

- Lacking comparison to using only PSM: I wonder if the authors considered using only PSM, instead of som combination of DSM and PSM. Is DSM necessary in high-t regime?

---

> ### Author Response · Authors · 2025-06-15
> **Response to Reviewer whEG**
>
> We sincerely thank the reviewer for the constructive feedback. Below we provide detailed responses, and revisions can be seen in \textcolor{red}{RED} in the revised manuscript.
>
> \paragraph{Novelty.}
> PSM presents an advancement in score matching for molecular systems, both in terms of application and methodological innovation.
>
> In contrast to TSM, which is limited to low-dimensional Gaussian settings and it relies on closed-form probability expressions, PSM is specifically designed for high-dimensional, structured molecular systems—a setting that has been identified as a major challenge in recent work, including iDEM [3]. It reinterprets log-density gradients as physically meaningful force-like labels, making it applicable to molecular modeling tasks beyond synthetic benchmarks. These systems involve intricate geometric structures and symmetries that standard MLPs, as used in TSM, fail to capture.
> PSM incorporates the Equiformer-v2 architecture with built-in equivariance and a novel time embedding mechanism, making it well-suited for structured, high-dimensional diffusion learning.
>
> We further provide new theoretical insights tailored to PSM. Specifically, we show that in the small-noise regime (i.e., small $t$), the optimal solution to the PSM objective more accurately approximates the true label under the Boltzmann distribution than DSM, and the variance of the PSM label decreases as time near $0$.
> These demonstrate PSM’s intrinsic debiasing capability and empirical effectiveness.
>
> \paragraph{Lack of Theoretical Guarantees to Generate Exact Samples from the Boltzmann Distribution.}
> Our label construction is directly grounded in the Boltzmann distribution $p \propto e^{-\frac{\mathcal{E}}{k_B T}}$, providing a principled basis for learning approximately Boltzmann-consistent behavior.
> Importantly, our goal is not to design a sequence generator that can produce exact samples from the Boltzmann distribution. Rather, we recognize that MD simulation is computationally expensive due to the need to generate long, ergodic trajectories. To address this, we train our model on short trajectory data, which has been empirically shown to exhibit bias relative to the true Boltzmann distribution.
> We formally show that PSM labels are closer to Boltzmann-consistent targets and exhibit lower variance in the small-$t$ regime (Lemma 2, Theorem 2). In this context, the term \emph{debiasing} refers to the correction of biases introduced by the short-trajectory training data.
> Empirically, ensemble-averaged observables $h(r)$ from PSM closely match those from unbiased MD simulations, indicating that PSM captures key physical statistics despite being trained on biased data.
>
> \paragraph{No Comparison with Molecular Dynamics (MD).}
> We compare PSM with molecular dynamics simulations on the aspirin molecule. For MD, we adopt the QuinNet force field [2], following the original training protocol.
> We measure both the force field training time and the time required to generate sufficiently long, converged trajectories (treated as the sampling time). For PSM, we train the diffusion model for $500$ epochs until convergence and measure the sampling time using the Euler discretization scheme. All experiments are conducted on a single NVIDIA A100 GPU.
> Our results show that PSM completes training in approximately $8$ hours and requires only $10$ minutes to generate $1,000$ batches, each consisting of $1,000$ time steps. In contrast, training the MD force field to a comparable accuracy takes around 24 hours, while achieving state-of-the-art performance may require up to one week of training. Moreover, the MD simulation phase takes about $31$ hours and $18$ minutes to generate $210,000$ steps—the trajectory length used in the reference MD17 aspirin dataset. These results demonstrate that PSM offers substantially improved sampling efficiency over MD.
>
> \paragraph{No Comparison with Using Only PSM.}
> We conduct experiments using PSM alone on 1D and 2D toy distributions (Figure 6, Appendix C), where $p(x) \propto e^{-5\|x\|^2 + \|x\|^4}$ and the force label is $(-4 \|x\|_2^2 + 10)x$. Results show that PSM and Piecewise perform comparably to DSM, and better capture the low-density central region.
> In complex molecular systems, however, PSM alone becomes less stable due to high label variance at large $t$ (Lemma 2, Appendix B.3). We also observe that force-based supervision becomes insufficient in high-dimensional cases, possibly because the projection of the force $\mathbf{F}$ onto the sample space is small, leading to weak learning signals.
> These findings motivate our hybrid design, combining PSM at small $t$ with DSM at large $t$ to improve stability and generalization.
>
> Reference:
>
> [2] QuinNet: Efficiently Incorporating Quintuple Interactions into Geometric Deep Learning Force Fields, Zun Wang et al.
>
> [3] Iterated Denoising Energy Matching for Sampling from Boltzmann Densities, Akhound-Sadegh et al.

---

### Review · Reviewer_oAeN · 2025-06-03

**Summary Of Contributions:**

This paper introduces Potential score matching,  a score-based diffusion model variant which is able to leverage knowledge about the target distribution to improve the behaviour and performance of the model at small times (i.e. when close to the data distribution).

The strategy is based on an alternative representation of the score-function in terms of the gradient of the log target density, which results in a loss function which requires access to the grad log target density at the data-points (thus can be considered additional label information).    Similar strategies have been explored before, e.g. in [Bortoli et al, 2024] targeted score matching paper, but the authors propose a novel way of integrating the psm score with dsm through interpolation.

This setting plays very well in the context of molecular dynamics, where you would typically have access to the energy and force values, as the training data would have arisen from MD simulations, etc.     To this end, the authors demonstrate the efficacy of their approaches (combining PSM with classical DSM) on a number of benchmark molecular problems, showing that their approach is effective.

**Audience:**

Yes

**Claims And Evidence:**

Yes

**Requested Changes:**

Can the authors please explain in more detail:
1. Why the need for piecewise and piecewise-weighted losses,  e.g. why not just use PSM all the way through.
2. Why the specific form for the piecewise-weighted loss, i.e. the use of t_dsm and t_psm are completely unclear to me.

The section defining these two needs to be substantially rewritten and clarified.

The authors should be a bit more explicit about going from the score-based approach to the denoiser approach -- why do this, what's the advantage in this setting?    Also, why and what the x_0 loss and epsilon loss, (5) and (6) respectively, needs to be explained in more detail,  for example, what is $\mathcal{D}\_\theta$ a function of?    They do cite Luo 2022, but it would be better to provide clarity directly within the text.   Similarly, jumping from denoiser to score is a bit 'sloppy' at times, for example in Page 2, it is not really correct to say "leveraging the relationship $s_{\theta}$ = ....    The relationship is between the losses - one can choose to optimise either score or denoiser loss, but not simply prescribe equality between the two networks.


Page 15: typo "Repressentation".

Lemma 1 and Theorem 3 -- these are largely undergraduate textbook proofs.   I would simply cite the relevant sources here, and dedicate the space to explaining the other aspects (denoising vs score based approaches in far more detail).

**Strengths And Weaknesses:**

Strengths:
1. The paper is nice because it recognizes the unique setting of molecular systems, and finds a bespoke diffusion strategy for it and exploits it effectively.
2. The numerical simulations are quite comprehensive, and clearly demonstrate the efficacy of the strategy.

Weaknesses:
1. The method itself is not really novel, considering [Bortoli, et al, 2024].    The authors do propose some variants of the approach which are novel, combining classical Diffusion Score Matching losses with the new PSM loss, but at no point is it really explained why such approaches are beneficial or even necessary.   This is the main weakness of the paper, in my opinion.
2. The paper has some issues with clarity and motivation in some areas - which I will detail in the next section.
3. The paper is a bit inconsistent in the mathematical exposition.   Some quite technical aspects (which are arguably well known in the diffusion literature) are completely glossed over (e.g. the relationship between the score and denoiser, etc), while very basic aspects of probability theory are expanded on in great detail.
4. The notation is challenging at times

---

> ### Author Response · Authors · 2025-06-15
> **Response to Reviewer oAeN**
>
> Thank you for your valuable feedback. We have marked all revisions in the revised manuscript in \textcolor{red}{RED} for clarity.
>
> \paragraph{Why Piecewise and Piecewise Weighted Losses Are Needed.}
> Previous studies [1, 4] have highlighted that DSM suffers from high variance and numerical instabilities near $t=0$. Building upon these findings, our theoretical analysis (Lemma 2 and Theorem 2) demonstrates that PSM significantly reduces variance and corrects estimation bias in the small-t regime, providing strong motivation for its use when $t$ is close to zero.
>
> PSM performs particularly well in low-dimensional settings, especially when the underlying energy function admits an analytical form. As shown in the toy model experiment (Appendix C, Figure 6), applying PSM across the entire time horizon achieves comparable accuracy and stability to DSM, while better capturing regions of low data density.
> However, in high-dimensional and complex molecular systems, our experiments show that force-based supervision alone becomes insufficient. Prior work [3, 4] has also noted this high-dimensional challenge. In our experiments, we further observe that the benefits of force-based guidance diminish as $t$ increases—likely due to the limited information retained from the initial dynamics at $t=0$ within the highly noisy $x_t$.
> These observations motivate our hybrid approach: applying PSM at small $t$ (i.e. $[0, t_{\text{p}}]$) to leverage its debiasing capability, and DSM (i.e. $[t_{\text{p}}, 1]$) elsewhere for a chosen $t_p$ to ensure stability. This design is both theoretically grounded and empirically validated. In our work, $t_{\text{psm}}$ refers to the denoising time used to construct the PSM loss, and similarly, $t_{\text{dsm}}$ refers to the time used for the DSM loss.
>
> In particular, the specific form for the Piecewise Weighted loss in the article is just a special case that we chose. In fact, it is sufficient to ensure that the weight of the PSM label is close to $1$ when $t$ is small, and the weight of the DSM is close to $1$ when $t$ is large.
>
> \paragraph{Novelty.} Although PSM shares a similar log-density gradient label structure with TSM, it introduces methodological and practical advancements.
>
> - \textbf{Application and architectural design:}
> In PSM, we reinterpret the log-density gradient labels as physical force fields for molecular system. More importantly, TSM has been validated only on low-dimensional Gaussian distributions and rely on the analytic expressions of energy. However, PSM is explicitly designed for high-dimensional molecular systems, which is addressed as a challenge in previous work [3, 4]. These systems exhibit intricate structures and symmetries. TSM relies on standard MLPs, which are inadequate for high-dimensional, structured data. PSM adopts the \texttt{Equiformer-v2} model with equivariance and a novel time embedding mechanism.
>
> - \textbf{Theoretical insight:} We theoretically show that for small $t$, the optimal solution for PSM network yields a more accurately approximates the label where samples satisfy the Boltzmann distribution than DSM. We also analyze the label variance for both DSM and PSM. These theories support the debiasing capability of PSM.
>
> \paragraph{Clarity and Notations.}
> We appreciate your suggestions regarding clarity and notation. We have updated the relevant sections accordingly in \textcolor{red}{RED} in the revised version.
>
> \paragraph{Mathematical Exposition and Background Material.} Thank you for highlighting the imbalance between technical depth and introductory explanations. The derivations of the $x_0$-loss and $\epsilon$-loss are provided in Equations (22) and (23) in Appendix B.1. Due to response length limitations, a more detailed discussion of the relationships among the $x_0$-loss, $s$-loss, and $\epsilon$-loss is presented in Appendix B.2.
>
> \textbf{Reference:}
>
> [1] Yang, Zhantao and Feng, Ruili and Zhang, Han and Shen, Yujun and Zhu, Kai and Huang, Lianghua, et. al. Lipschitz Singularities in Diffusion Models.
>
> [3] Iterated Denoising Energy Matching for Sampling from Boltzmann Densities, Akhound-Sadegh, Tara and Rector-Brooks, Jarrid and Bose, Avishek Joey and Mittal, Sarthak and Lemos, Pablo and Liu, Cheng-Hao and Sendera, Marcin and Ravanbakhsh, Siamak and Gidel, Gauthier and Bengio, Yoshua and others.
>
> [4] Target Score Matching, Valentin De Bortoli and Michael Hutchinson and Peter Wirnsberger and Arnaud Doucet.

---

### Decision · Action_Editor_WhHh · 2025-07-21

**Recommendation:** Accept as is

**Additional Comments:**

Not strictly a revision request, but building on the reviews, I would encourage the authors to empirically differentiate TSM & PSM further, since they are closely related.

**Audience:**

Yes

**Audience Explanation:**

State-of-the-art work on molecular Boltzmann samplers demonstrates how hard this problem still is. This paper is an interesting contribution to that subfield.

**Claims And Evidence:**

Yes

**Claims Explanation:**

Although the reviewers are unanimous in their assessment that the novelty of this paper is limited, they all agree that the work is correct, and its claims are supported by accurate, convincing and clear evidence.

---

> ### Author Response · Authors · 2025-07-28
> **Response to Action Editor WhHh**
>
> Thank you very much for the constructive feedback and for recommending the acceptance of our manuscript.
> While our work shares a conceptual connection with TSM, PSM extends its applicability and methodology.
>
> TSM operates under low-dimensional Gaussian assumptions with access to closed-form densities, whereas PSM targets high-dimensional, structured molecular systems—a setting known to be challenging, as highlighted in recent work like iDEM.
> To address this, PSM reinterprets log-density gradients as physically meaningful, force-like labels, making it suitable for realistic molecular modeling beyond synthetic benchmarks. Unlike TSM’s MLP-based architecture, PSM leverages the Equiformer-v2 backbone with built-in equivariance and a novel time embedding mechanism to capture geometric symmetries essential in molecular systems.
> Furthermore, we provide new theoretical insights: in the small-noise regime ($t \to 0$), the PSM objective leads to a more accurate and lower-variance approximation to the true force under the Boltzmann distribution than DSM, demonstrating PSM’s intrinsic debiasing property.
>
> We sincerely appreciate your time, thoughtful comments, and support of our submission. We will incorporate this clarification into the final version before the camera-ready deadline.